# Genome-wide analysis of Smad and Schnurri transcription factors in *C. elegans* demonstrates widespread interaction and a function in collagen secretion

Mehul Vora[1,2†], Jonathan Dietz[1†], Zachary Wing[3], Karen George[1], Jun Kelly Liu[4], Christopher Rongo[1]*, Cathy Savage-Dunn[3,5]*

[1]Waksman Institute, Department of Genetics, Rutgers University, New Brunswick, United States; [2]ModOmics Ltd, Southampton, United Kingdom; [3]Department of Biology, Queens College, CUNY, New York, United States; [4]Department of Molecular Biology and Genetics, Cornell University, Ithaca, United States; [5]PhD Program in Biology, The Graduate Center, CUNY, New York, United States

*For correspondence:
crongo@waksman.rutgers.edu
(CR);
cathy.savagedunn@qc.cuny.edu
(CS-D)

†These authors contributed equally to this work

## eLife Assessment

Modulation of BMP signaling affects body size in the nematode *Caenorhabditis elegans*, and this paper examines the effects on *C. elegans* body size brought about by the modulation of BMP signaling. The study provides **valuable** analyses of ChIP-seq and RNA-seq data to understand the function of SMA-3 (Smad) and SMA-9 (Schnurri) in this model. The authors provide **compelling** evidence that the BMP-dependent body size effect could be due to defects in cuticle collagen secretion, a finding of interest to those studying organismal growth and epidermal function.

**Abstract** Smads and their transcription factor partners mediate the transcriptional responses of target cells to secreted ligands of the transforming growth factor-β (TGF-β) family, including those of the conserved bone morphogenetic protein (BMP) family, yet only a small number of direct target genes have been well characterized. In *C. elegans*, the BMP2/4 ortholog DBL-1 regulates multiple biological functions, including body size, via a canonical receptor-Smad signaling cascade. Here, we identify functional binding sites for SMA-3/Smad and its transcriptional partner SMA-9/Schnurri based on ChIP-seq peaks (identified by modEncode) and expression differences of nearby genes identified from RNA-seq analysis of corresponding mutants. We found that SMA-3 and SMA-9 have both overlapping and unique target genes. At a genome-wide scale, SMA-3/Smad acts as a transcriptional activator, whereas SMA-9/Schnurri direct targets include both activated and repressed genes. Mutations in *sma-9* partially suppress the small body size phenotype of *sma-3*, suggesting some level of antagonism between these factors and challenging the prevailing model for Schnurri function. Functional analysis of target genes revealed a novel role in body size for genes involved in one-carbon metabolism and in the endoplasmic reticulum (ER) secretory pathway, including the disulfide reductase *dpy-11*. Our findings indicate that Smads and SMA-9/Schnurri have previously unappreciated complex genetic and genomic regulatory interactions that in turn regulate the secretion of extracellular components like collagen into the cuticle to mediate body size regulation.

**eLife digest** Growth and development depend on the ability of cells to communicate through an intricate ballet of molecular signals that determine cell behaviors. Signaling proteins belonging to the BMP family, in particular, play an important role by activating certain 'transcription factors', known as Smad proteins, which then bind to specific DNA sequences to switch target genes on or off. The activity of these BMP-activated transcription factors is modulated by other molecular partners – Schnurri, for instance, is a well-known transcription factor partner of Smad. Still, exactly how Smad and Schnurri interact to control the expression of genes across the genome, and thereby execute complex biological programs, has remained unclear.

To investigate this question, Vora, Dietz et al. examined how Smad and Schnurri partner to regulate body size during the development of *Caenorhabditis elegans*, a transparent, non-parasitic worm widely used in research. Various genomic techniques were used to reveal where on the genome the transcription factors could bind, as well as to track resulting changes in gene expression. Software approaches were then adapted to combine these datasets, showing that Smad and Schnurri have both shared and independent targets. The two proteins usually cooperate to activate gene expression, but they sometimes antagonize each other's functions. Finally, analyses showed that Smad and Schnurri help control body size by regulating the secretion of collagen. This protein is the primary component of the cuticle, a flexible external layer that shields the worms from the environment as well as determines their bodies' shape and size. Overall, the work by Vora, Dietz et al. demonstrates that previous models for Smad and Schnurri interactions were incomplete, paving the way for further research into these proteins and their role in development.

## Introduction

Members of the TGF-β family of secreted ligands play numerous roles in development and disease. In humans, there are 33 ligand genes that can be broadly separated into two subfamilies: the TGF-β/Activin subfamily and the BMP subfamily (*Massagué, 1998*). Due to the conservation of these ligands and their signaling pathways across metazoans, genetic studies in invertebrate systems have been instrumental in identifying signaling mechanisms (*Savage et al., 1996*; *Sekelsky et al., 1995*). Canonical signaling occurs when ligand dimers bind to transmembrane receptors generating a heterotetrameric complex consisting of two type I and two type II serine/threonine kinase receptors. Following ligand binding and complex assembly, the constitutively active type II receptor phosphorylates the type I receptor on the GS domain and thereby activates its kinase domain (*Wrana et al., 1994*). The activated type I receptor phosphorylates the C-terminus of intracellular receptor-regulated Smads (R-Smads), promoting their heterotrimeric complex formation with co-Smads. The heterotrimeric Smad complex accumulates in the nucleus and binds DNA directly to elicit changes in gene expression (*Liu et al., 1995*; *Wrana et al., 1992*; *Estevez et al., 1993*; *Gerstein et al., 2010*; *Chacko et al., 2004*; *Souchelnytskyi et al., 1997*; *Qin et al., 2001*; *Zhang et al., 1996*; *Abdollah et al., 1997*; *Lagna et al., 1996*; *Nicolás et al., 2004*). Co-Smads for all ligands and R-Smads for TGF-β/Activin ligands bind a 4 bp GTCT Smad Binding Element (SBE); furthermore, R-Smads for BMP ligands associate with GC-rich sequences (GC-SBE) (*Kim et al., 1997*; *Gao et al., 2005*; *Rushlow et al., 2001*). The SBE is considered too degenerate and low affinity to account fully for binding specificity, so transcription factor partners likely contribute to target gene selection (*Hill, 2016*). To date, only a few direct target genes of Smads have been extensively studied, including *Drosophila brinker* (*Pyrowolakis et al., 2004*); Xenopus *mixer* (*Germain et al., 2000*) and *Xvent2* (*Yao et al., 2006*); and the mammalian *ATF3* and *Id* genes (*Kang et al., 2003*). Genome-wide studies have the potential to expand these examples and elucidate general principles of target gene selection (*Deignan et al., 2016*; *Chiu et al., 2014*; *Morikawa et al., 2011*). More of these studies are needed to understand how Smad transcriptional partners influence target gene selection and contribute to the execution of specific biological functions.

In the nematode *Caenorhabditis elegans,* a BMP signaling cascade initiated by the ligand DBL-1 plays a major role in body size regulation (*Suzuki et al., 1999*). In nematodes, body size is constrained by a collagen-rich cuticle, which is secreted by an epidermal layer (the hypodermis) and remodeled over four successive molts during larval growth and then continuously during adulthood (*Lazetic*

*and Fay, 2017*; *Page and Johnstone, 2007*). DBL-1 signals through type I receptor SMA-6, type II receptor DAF-4, and Smads SMA-2, SMA-3, and SMA-4 (founding members of the Smad family), which act together in the hypodermis to promote body size growth during the earliest larval growth stages (*Gumienny and Savage-Dunn, 2013*; *Yoshida et al., 2001*; *Wang et al., 2002*). The exact mechanism by which the DBL-1 pathway regulates body size is not fully understood, but is known to involve the regulated synthesis of cuticular collagen, of which there are over 170 genes (*Roberts et al., 2010*; *Liang et al., 2007*; *Madaan et al., 2018*). A complete understanding of how DBL-1 regulates body size will require the identification of all direct transcriptional targets of the pathway during larval growth.

In addition to body size, the DBL-1 pathway also regulates male tail patterning, mesodermal lineage specification, innate immunity, and lipid metabolism. A transcription factor partner for this pathway, SMA-9, has been identified that plays a role in each of these biological functions (*Foehr et al., 2006*; *Liang et al., 2003*). Unlike for the core components of the signaling pathway, however, loss of SMA-9 function can result in a different effect depending on the phenotype, suggesting that this factor can either be an equivalent co-factor, a factor with a more limited role, or an antagonistic factor depending on the specific function (*Foehr et al., 2006*; *Liang et al., 2003*; *Clark et al., 2018a*). SMA-9 is the homolog of *Drosophila* Schnurri, which was identified for its roles in Dpp/BMP signaling (*Staehling-Hampton et al., 1995*; *Grieder et al., 1995*; *Arora et al., 1995*). Three vertebrate Schnurri homologs regulate immunity, adipogenesis, and skeletogenesis, acting through both BMP-dependent and BMP-independent mechanisms (*Jin et al., 2006*; *Jones and Glimcher, 2010*; *Jones et al., 2006*; *Steinfeld et al., 2016*). Schnurri proteins are very large transcription factors with multiple Zn-finger domains. At the *brinker* locus in *Drosophila* and the *Xvent2* locus in Xenopus, binding of an R-Smad and a Co-Smad with a precise 5 bp spacing between binding sites has been shown to recruit Schnurri, which controls the direction of transcriptional regulation (*Yao et al., 2006*). This model for Smad-Schnurri interaction has not been tested at a genomic scale.

In this study, we use BETA software to combine RNA-seq and ChIP-seq datasets for SMA-3/Smad and SMA-9/Schnurri to identify direct versus indirect target genes of these factors, as well as to identify common versus unique targets (*Vora et al., 2022*; *Wang et al., 2013*). Analysis of *sma-3; sma-9* double mutants further extends our understanding of how these factors interact to produce locus-specific effects on target genes. We use GO term analysis and loss-of-function studies that shed light on the downstream effectors for body size regulation, lipid metabolism, and innate immunity. Finally, we use a ROL-6::wrmScarlet reporter for collagen synthesis and secretion to show that SMA-3, SMA-9, and the transcriptional target gene DPY-11 regulate body size growth by promoting the secretion and delivery of collagen into the cuticle.

## Results

### Transcription factors SMA-3 and SMA-9 bind overlapping and distinct genomic sites

Smads and Schnurri are known to bind DNA as a physical complex (*Dai et al., 2000*; *Müller et al., 2003*), but a limitation of the previous work is that only a small number of specific target genes were analyzed. We sought to determine the extent to which these factors act together or independently by identifying the binding sites of SMA-3 and SMA-9 on a genome-wide scale. We generated GFP-tagged transgenes for SMA-3 and SMA-9, and then demonstrated that they are functional, as evidenced by their ability to rescue the mutant phenotypes of respective loss-of-function *sma-3* and *sma-9* mutants (*Foehr et al., 2006*; *Clark et al., 2018a*). We provided these constructs to the modEN-CODE/modERN consortium, which then analyzed genome binding via ChIP-seq at the second larval (L2) stage, a developmental stage at which a Smad reporter is highly active, and both SMA-3 and SMA-9 are first observed to affect body size (*Kudron et al., 2018*; *Tian et al., 2010*). ChIP sequencing reads identified 4205 peaks for GFP::SMA-3 and 7065 peaks for SMA-9::GFP (*Supplementary file 1*). Although these data were previously released publicly as part of the modENCODE/modERN consortium (*Kudron et al., 2018*; *Gerstein et al., 2010*), our examination here provides the first comprehensive analysis of these datasets.

Because Smads and Schnurri are known to form a complex on DNA, we sought to determine the frequency with which SMA-3 and SMA-9 bind together at a genome-wide scale by analyzing the

distances between the centroids of SMA-3 and SMA-9 ChIP-seq peaks. If SMA-3 and SMA-9 bind independently, then we would expect a Gaussian distribution of inter-centroid distances, whereas if they frequently bind in a complex, we should see a non-Gaussian distribution with increased representation of distances less than or equal to the average peak size. This analysis demonstrated an increased representation (approaching 45%) of inter-centroid distances of 100 bp or less (*Figure 1a and b*), smaller than the average peak size for SMA-3 (400 bp) or SMA-9 (250 bp) (*Figure 1c*), consistent with the interpretation that SMA-3 and SMA-9 frequently bind as either subunits in a complex or in close vicinity to each other along the DNA. The midpoint of the cumulative probability distribution of inter-centroid distances was 788 bp; by contrast, a randomization of the positions of SMA-3 and SMA-9 ChIP peaks expanded the midpoint of the cumulative probability distribution of inter-centroid distances to 8211 bp (*Figure 1a and b*). From this analysis, a substantial subset (3101 peaks) of the SMA-3 (73.7%) and SMA-9 (43.9%) peaks overlap (*Figure 1d*). We, therefore, considered these instances of overlapping peaks to be evidence of SMA-3/SMA-9 association (possibly physical complexes, although this would require a formal biochemical demonstration), whereas adjacent but non-overlapping peaks likely represent an independent binding, leading us to conclude that (1) SMA-3 typically binds together with SMA-9 to DNA sites, and that (2) over half of all SMA-9 sites do not overlap with these SMA-3 occupied sites.

## Identification of direct transcriptional targets of SMA-3 and SMA-9

To determine how these binding sites correlate with changes in gene expression of neighboring genes, we performed RNA-seq on L2 stage samples of *sma-3* and *sma-9* mutants compared with wild-type controls. Principal component analysis (PCA) demonstrated that all three biologically independent replicates of each genotype clustered together (*Figure 2a*) and that each genotype is transcriptionally distinct from the others. Using a false discovery rate (FDR)≤0.05, we identified 1093 differentially expressed genes (DEGs) downregulated and 774 upregulated DEGs in *sma-3* mutants (*Figure 2b*, *Supplementary file 2*). In *sma-9* mutants, we identified 412 downregulated DEGs and 371 upregulated DEGs (*Figure 2d*, *Supplementary file 2*). Previously identified target genes, such as *fat-6* and *zip-10* (*Liang et al., 2007*), were also found in these datasets, confirming the effectiveness of the RNA-seq experiments.

RNA-seq identifies both direct and indirect transcriptional targets. To identify direct functional targets of each of these transcription factors, we employed BETA software (*Figure 2c and e*), which infers direct target genes by integrating ChIP-seq and RNA-seq data (*Wang et al., 2013*). The BETA analysis identified 367 direct targets for SMA-3 and 332 direct targets for SMA-9 (*Supplementary file 3*). Every identified direct target of SMA-3 was downregulated in the *sma-3* mutant (*Figure 2b*), indicating that SMA-3/Smad functions primarily as a transcriptional activator. In contrast, 46% of direct targets of SMA-9 were upregulated and 53% were downregulated in the *sma-9* mutant (*Figure 2d*, *Supplementary file 3*). Thus, SMA-9 likely acts as either a transcriptional activator or repressor depending on the genomic context. This conclusion is consistent with our previous analyses of SMA-9 function in vivo and in a heterologous system (*Liang et al., 2007*).

## Significant overlap in directly regulated DEGs of SMA-3 and SMA-9

Because SMA-3 and SMA-9 ChIP-seq peaks often overlapped along the DNA, we sought to identify a core subset of DEGs co-regulated by these transcription factors. Rather than relying on individual RNA-seq analyses, in which arbitrary cut-offs for significance may lead to an underestimation of the overlap, we performed Luperchio Overlap Analysis (LOA) on *sma-3* and *sma-9* RNA-seq datasets to identify 882 shared DEGs (*Supplementary file 4*, *Vora et al., 2022*; *Luperchio et al., 2021*). From ChIP-seq data, we identified 3101 peaks that are overlapping between SMA-3 and SMA-9. We used LOA to identify DEGs shared between the pairwise comparison of *sma-3* versus wild-type, and between the pairwise comparison of *sma-9* versus wild-type, using evidence of potential DEGs in one comparison to inform the state of those potential DEGs in the other comparison. Processing common occupancy sites with the common DEGs through BETA software (*Figure 3a*), we identified 129 co-regulated direct target genes (i.e. target genes showing differential expression in both *sma-3* and *sma-9* mutants versus wild-type, and with overlapping SMA-3 and SMA-9 binding peaks nearby (*Supplementary file 5*)). These results are consistent with SMA-3 and SMA-9 acting either in a protein complex or working together in close association along the DNA. Most (114) of these 129 co-regulated direct targets are

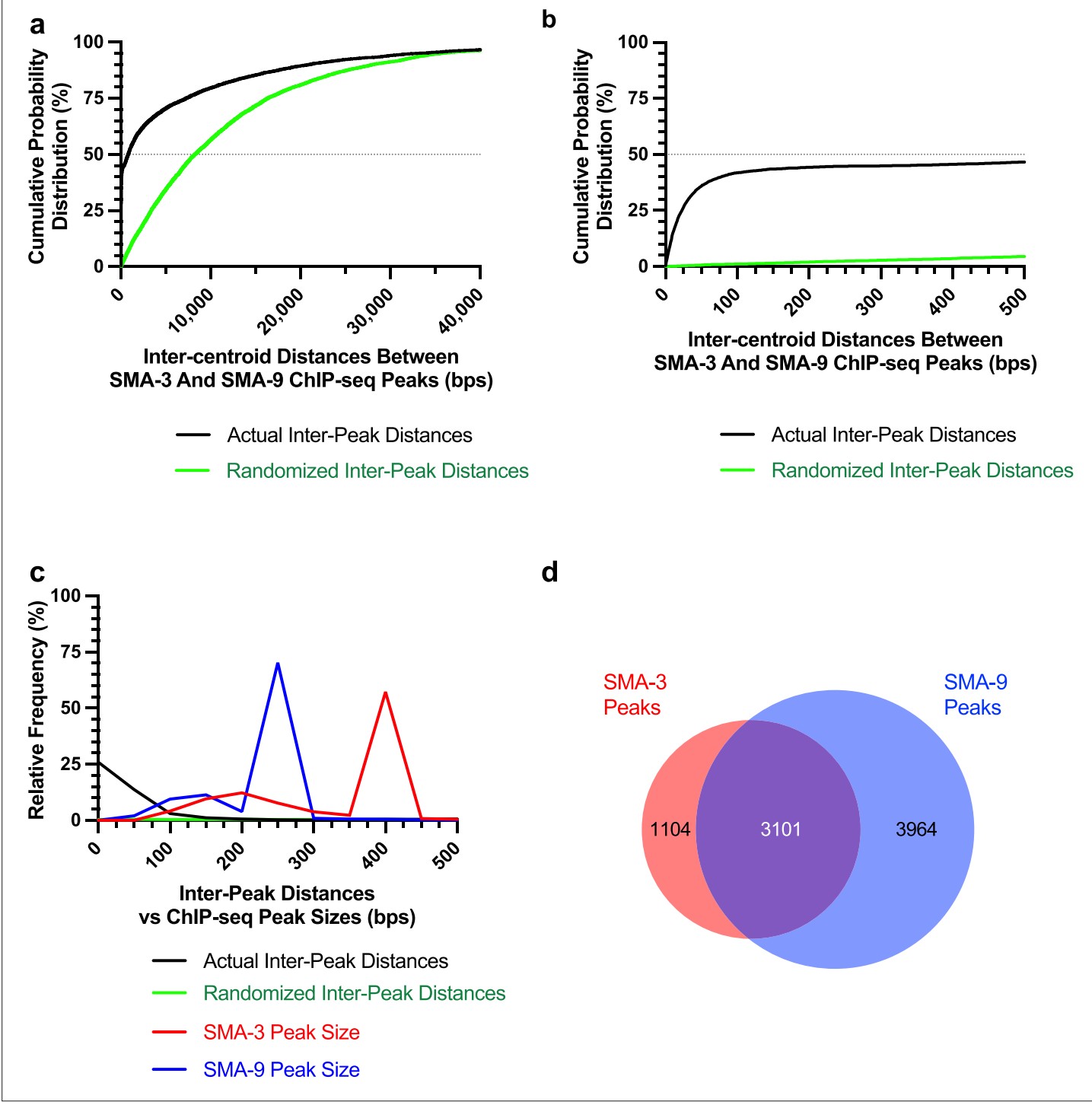

**Figure 1.** Transcription factors SMA-3 and SMA-9 bind both overlapping and distinct genomic sites. (**a**) Cumulative probability distribution graph of the distances between the centroids (base pair position located centrally within each peak) of nearest neighbor SMA-3 and SMA-9 Chromatin immunoprecipitation sequencing (ChIP-seq) peaks. The black line represents actual inter-peak distances, whereas the green line represents a hypothetical randomized dataset. The horizontal dotted line indicates the point in the curve at which half of the peak pairs fall. (**b**) Same cumulative probability distribution as in (**a**), but focused on distances less than 500 bps. The centroids of nearly half of all SMA-3/SMA-9 neighboring pairs fall within 500 bps of each other. (**c**) Histogram of the interpeak distances (actual data in black, randomized data in green), as well as the ChIP-seq peak widths (SMA-3 in red, SMA-9 in blue). Most peaks are larger in size than most interpeak centroid distances, indicating substantial peak overlap. (**d**) Size-proportional Venn diagram showing the number of SMA-3 and SMA-9 peaks that either overlap with one another or remain independent. Although most SMA-3 peaks overlap with SMA-9 peaks, more than half of SMA-9 peaks are located independently of SMA-3 peaks.

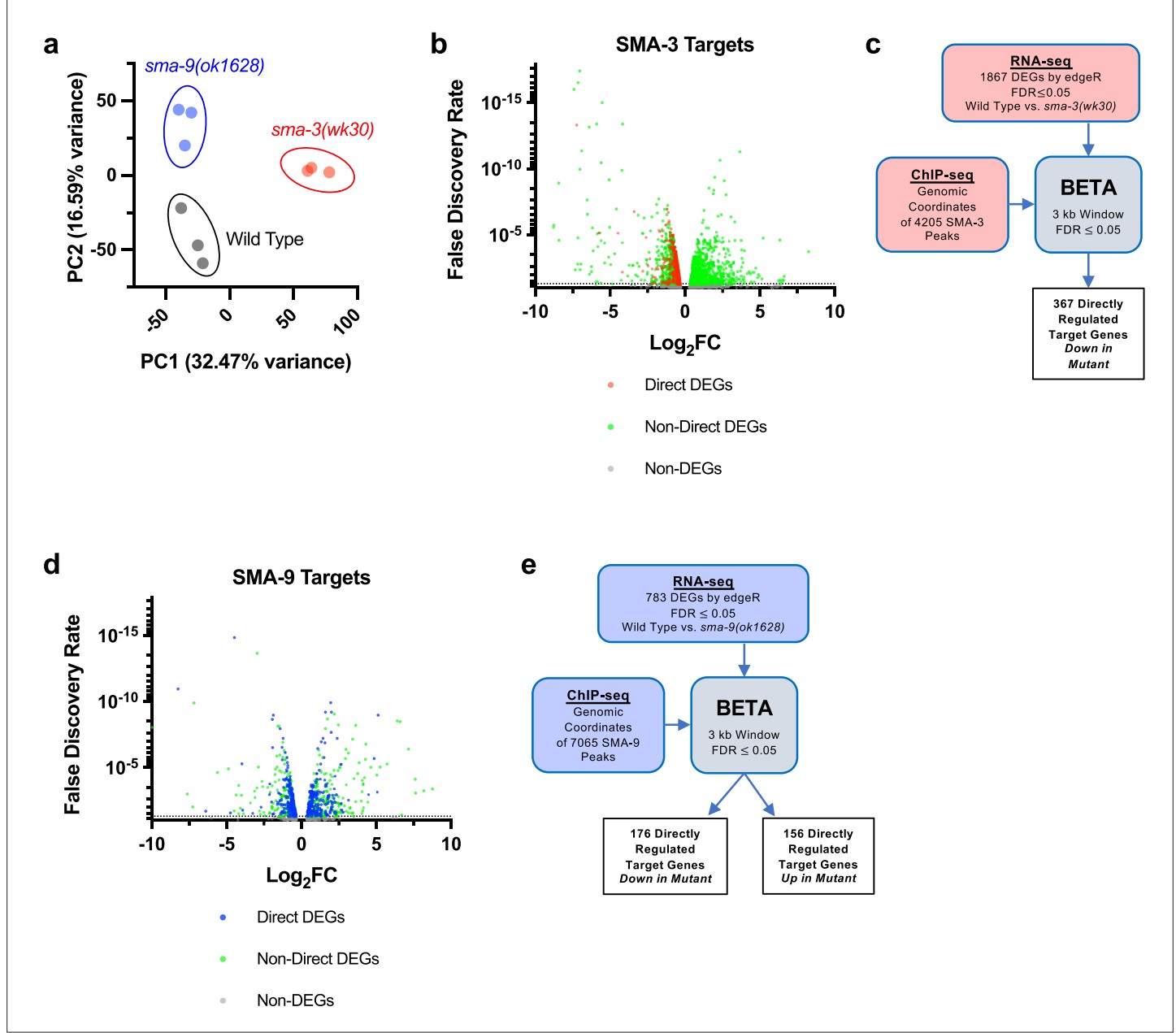

**Figure 2.** Identification of direct transcriptional targets of SMA-3 and SMA-9. (**a**) Principal component analysis (PCA) over two dimensions (PC1 and PC2) for RNA-seq datasets for wild-type (in black), *sma-3(wk30)* (in red), and *sma-9(ok1628)* (in blue). The percent of variance for each component is indicated. The three biological replicates for each genotype are well clustered. (**b**) Volcano plot of RNA-seq false discovery rate (FDR) values versus log2 fold change (FC) expression for individual genes (squares) in *sma-3* mutants relative to wild-type. The direct targets identified by BETA are indicated with red squares; the negative log2 FC values demonstrate that SMA-3 promotes the expression of these genes. Non-direct target genes nevertheless showing differential expression are indicated with green squares. (**c**) Strategy for integrating SMA-3 Chromatin immunoprecipitation sequencing (ChIP-seq) and mutant RNA-seq data to identify directly regulated targets. (**d**) Volcano plot of RNA-seq FDR values versus log2 fold change expression for individual genes (squares) in *sma-9* mutants relative to wild-type. The direct targets identified by BETA are indicated with blue squares; the combination of positive and negative log2FC values demonstrates that SMA-9 promotes the expression of some of these genes and inhibits the expression of others. Non-direct target genes nevertheless showing differential expression are indicated with green squares. (**e**) Strategy for integrating SMA-9 ChIP-seq and mutant RNA-seq data to identify directly regulated targets.

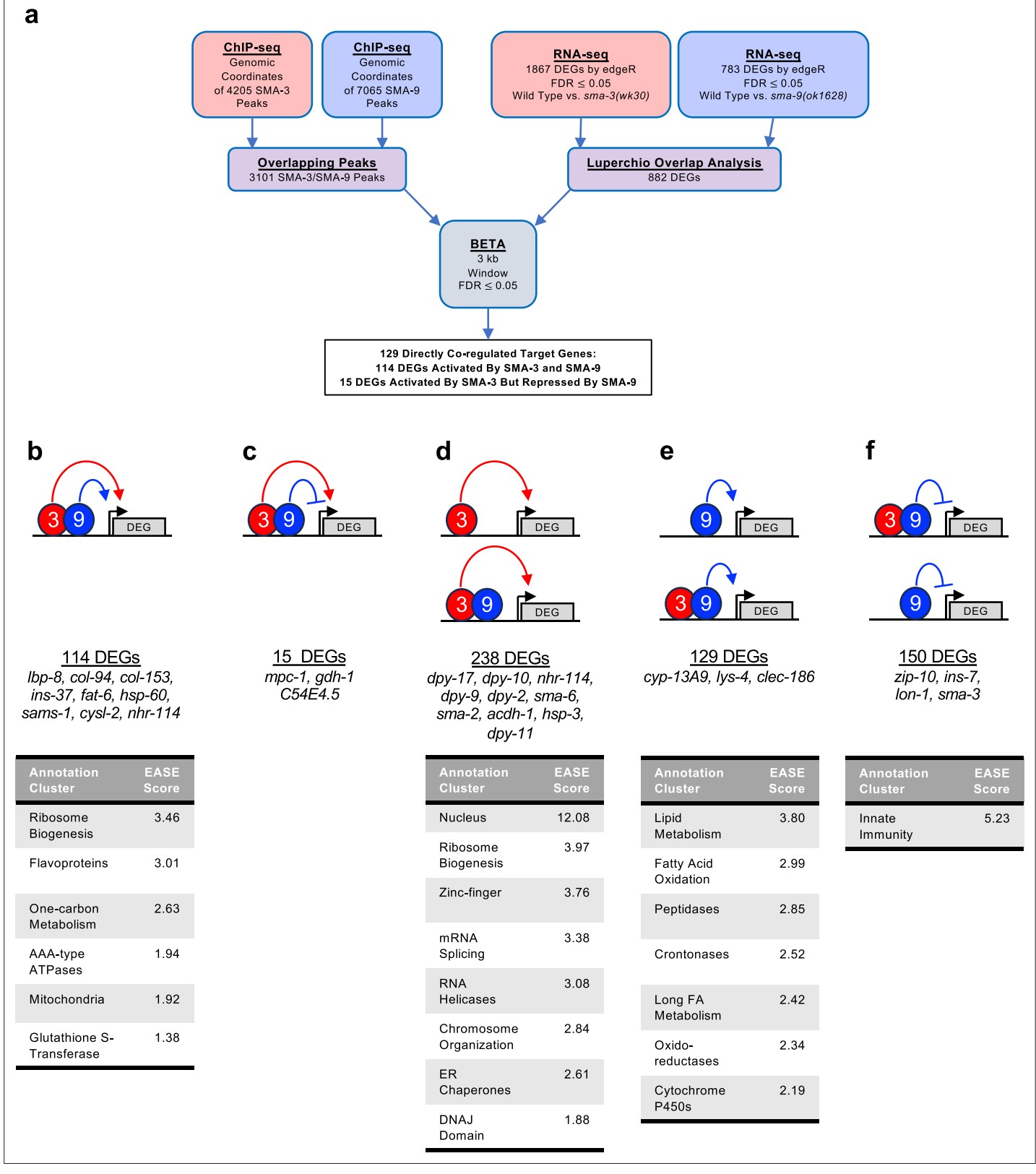

**Figure 3.** Significant overlap in directly regulated differentially expressed genes (DEGs) of SMA-3 and SMA-9. (**a**) Strategy for integrating SMA-3 and SMA-9 ChIP-seq and mutant RNA-seq data to identify common versus unique directly regulated targets. (**b–f**) Cartoon representations of the different types of direct target genes, their neighboring SMA-3 and/or SMA-9 binding sites, and the effect of those sites on that gene's expression. The red circle labeled '3' represents SMA-3 binding sites, whereas the blue circle labeled '9' represents SMA-9 binding sites. Arrows represent that the wild-type

*Figure 3 continued on next page*

*Figure 3 continued*

transcription factor promotes the expression of the neighboring DEG (gray), whereas T-bars indicate that it inhibits the expression of the DEG. Types of regulation include (**b**) SMA-3 alone promoting DEG expression, (**c**) SMA-3 and SMA-9 combined promoting expression, (**d**) SMA-3 and SMA-9 showing antagonistic regulation of expression, (**e**) SMA-9 alone promoting DEG expression, and (**f**) SMA-9 alone inhibiting DEG expression. Example DEGs and tables of annotation clusters for gene ontology terms for those DEGs (via DAVID, with accompanying statistical EASE score) are shown under the cartoon demonstrating each type of regulation.

The online version of this article includes the following figure supplement(s) for figure 3:

**Figure supplement 1.** DAVID annotation clusters of gene ontology terms for identified SMA-3 and SMA-9 targets.

**Figure supplement 2.** SMA-3 and SMA-9 sites tend not to be at HOT sites.

activated by both SMA-3 and SMA-9 (*Figure 3b*), but 15 of them have reversed regulation in *sma-9* mutants compared with *sma-3* (*Figure 3c*), suggesting an antagonistic function that we analyze further below. For shared activated targets, the loss of SMA-3 caused a greater fold change than the loss of SMA-9 (*Supplementary file 2*).

Our BETA/LOA analysis next allowed us to deduce SMA-3-exclusive and SMA-9-exclusive direct targets (i.e. target genes directly regulated exclusively by one factor or the other, but not both), revealing 238 SMA-3-exclusive direct targets (*Figure 3d*) and 279 SMA-9-exclusive direct targets (*Figure 3e and f*). About half (129) of the 279 SMA-9-exclusive direct targets are activated by SMA-9 (*Figure 3e*), whereas the other half (150) appear to be inhibited by SMA-9 (*Figure 3f*). Surprisingly, many of these target genes contained overlapping SMA-3 and SMA-9 binding peaks (*Figure 3b, e and f*), although a loss of one of the two factors did not result in changes in gene expression, perhaps suggesting that the presence of the other factor at these targets was sufficient to regulate gene expression to physiological levels. Interpretation is further complicated for target genes surrounded by a mixture of distinct and overlapping SMA-3 and SMA-9 peaks. Nevertheless, our results suggest that (1) SMA-3 and SMA-9 can act independently of one another, (2) they usually (although not always) act as transcriptional activators of shared target genes when they co-occupy the same sites along the DNA, and (3) SMA-9 without SMA-3 can act as either a transcriptional activator or repressor of its own SMA-3-independent targets.

The large number of total SMA-3 and SMA-9 ChIP-seq peaks yet the small number of functional peaks identified by BETA was surprising. The ModENCODE/ModERN consortium previously demonstrated the existence of High Occupancy Target (HOT) sites where ChIP-seq association with 15 or more transcription factors occurs, perhaps due to 'sticky' regions of the genome leading to false signals (*Kudron et al., 2018*). We surveyed the binding sites of transcription factors within the ModERN database for overlap with either SMA-3 or SMA-9 sites (*Figure 3—figure supplement 2*). Less than 25% of all the numerous SMA-3 or SMA-9 sites were associated with HOT sites. Restricting our analysis to just functional SMA-9 sites identified by BETA reduced HOT site association down to 15%, although a similar effect was not observed for functional SMA-3 sites identified by BETA. SMA-9-exclusive sites showed even lower HOT site association, whereas 40% of SMA-9 sites overlapping with SMA-3 sites (i.e. co-regulated) were associated with HOT sites. These results suggest that the large number of total SMA-3 and SMA-9 sites is not explained by binding to HOT sites, although there might be some affinity for SMA-3/SMA-9 co-regulated genes to be near HOT sites.

## Integration of SMA-3 and SMA-9 function

SMA-3 and SMA-9 both regulate body size, and a loss of function mutation in either gene results in a small body size phenotype (*Savage et al., 1996*; *Foehr et al., 2006*; *Liang et al., 2003*). We used double mutant analysis to determine whether SMA-3 and SMA-9 regulate body size independently or together. For two gene products that act together in the same pathway, we expect the double mutants to resemble one of the single mutants. If they function independently, then we expect the double mutant to be more severe than the single mutants (i.e. additive phenotypes). We constructed a *sma-3; sma-9* double mutant and measured its body length at the L4 stage in comparison with a wild-type control, *sma-3* mutants, and *sma-9* mutants. Contrary to expectations, the double mutant was neither the same as nor more severe than the single mutants; instead, it showed an intermediate phenotype (*Figure 4a*). This result suggests that SMA-9 may act as both a positive and negative regulator of body size, indicating some antagonistic activity towards SMA-3. One mechanism for this antagonism could be the repression of SMA-3 target genes by SMA-9. Alternatively, SMA-9-exclusive

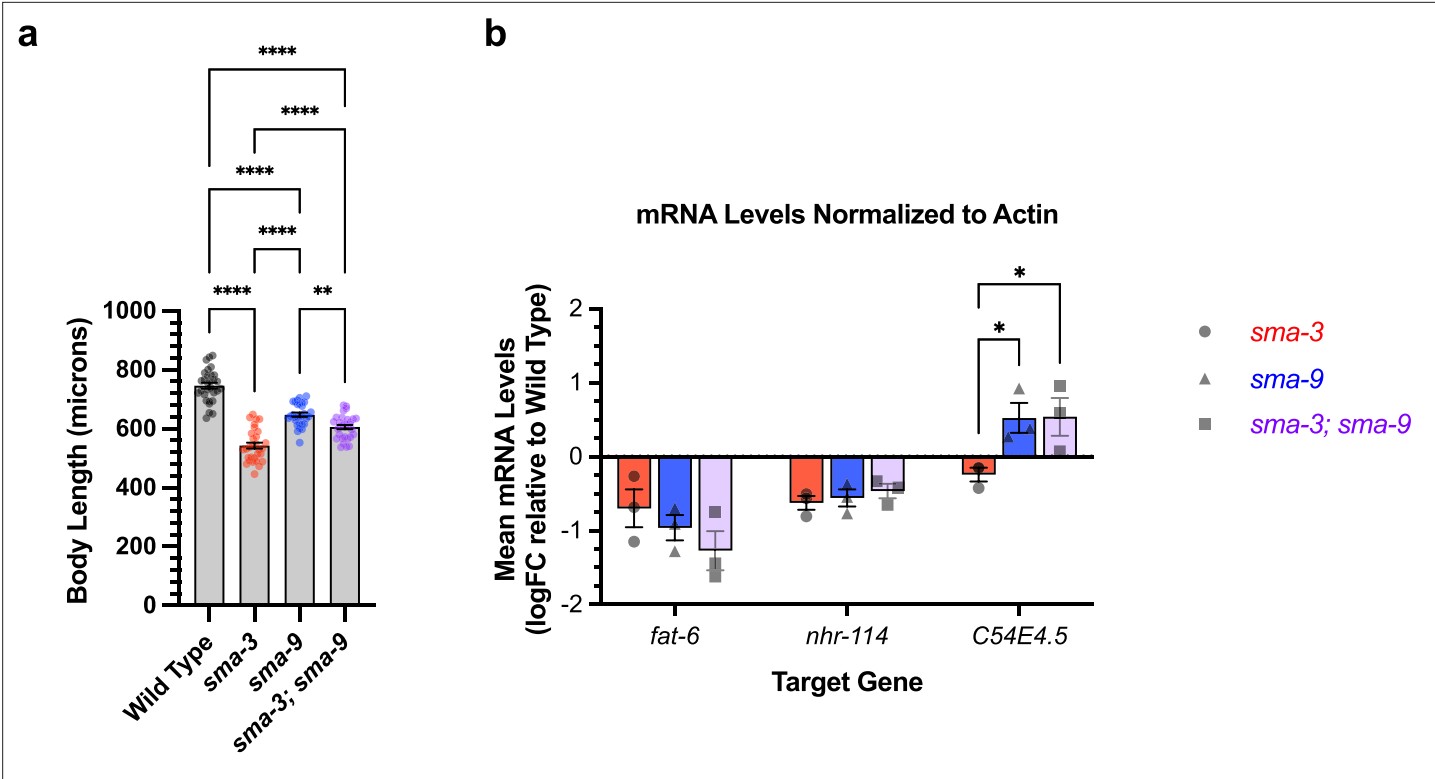

**Figure 4.** Genetic interactions between SMA-3 and SMA-9. (**a**) Mean body length of L4 animals (measured head to tail in microns). Dots indicate the size of individual animals. ****p<0.0001, **p<0.01 One-way ANOVA with Tukey's multiple comparison test. (**b**) Mean mRNA levels for the indicated target gene (x-axis) for the indicated genotypes (*sma-3*, *sma-9*, or the double mutant combination) relative to the level in the wild-type. Expression values are in log2 fold change (FC). Individual genotype mRNA levels for each gene were first normalized to actin mRNA levels in that genotype. *p<0.05 Two-way ANOVA with Tukey's multiple comparison test. Data was analyzed from three biological replicates.

The online version of this article includes the following figure supplement(s) for figure 4:

**Figure supplement 1.** Genes antagonistically regulated by SMA-3 and SMA-9.

target genes could negatively regulate body size. There is precedent for a BMP pathway component to have dual, opposite roles in body size regulation (***DeGroot et al., 2023***).

We hypothesized that the interaction between SMA-3 and SMA-9 may be context-dependent, with different target genes showing independent, coordinated, or antagonistic interactions between the transcription factors. We tested this hypothesis by performing qRT-PCR on select target genes in wild-type, *sma-3* mutants, *sma-9* mutants, and *sma-3; sma-9* double mutants at the L2 stage. We first considered three target genes: *fat-6*, which has overlapping SMA-3 and SMA-9 peaks and is downregulated in both *sma-3* and *sma-9* mutants; *nhr-114*, which has both overlapping and distinct non-overlapping SMA-3 and SMA-9 peaks and is downregulated in both *sma-3* and *sma-9* mutants; and *C54E4.5*, which has overlapping SMA-3 and SMA-9 peaks yet shows the opposite direction of regulation in *sma-3* versus *sma-9* mutants. For two of the tested target genes, *fat-6* and *nhr-114,* there was no significant difference in expression levels between the single and double mutants (***Figure 4b***), consistent with the transcription factors acting together. The third target gene, *C54E4.5*, was selected because it is a co-regulated direct target yet the RNA-seq data shows changes in its expression in opposite directions in *sma-3* versus *sma-9* mutants, downregulated in *sma-3* yet upregulated in *sma-9* relative to wild-type. In the double mutant, *C54E4.5* is upregulated and indistinguishable from the expression in *sma-9* single mutants (***Figure 4b***). Thus, for this target gene, the *sma-9* loss-of-function phenotype is epistatic to that of *sma-3*, indicating that SMA-9 is required for SMA-3 to regulate its expression.

We wanted to know whether this interaction occurred more generally, so we analyzed an additional five target genes that are regulated in opposite directions by SMA-3 and SMA-9: *arrd-19*, *nspe-7*, *nspc-16*, *catp-3*, and *gdh-1* (***Figure 4—figure supplement 1***). For each of these five genes, the

direction of regulation was confirmed by RT-PCR, although only two of them reached statistical significance. Furthermore, for all five genes, as for *C54E4.5*, expression in the *sma-3; sma-9* double mutant was not significantly different from that in the *sma-9* single mutant. Thus, for multiple target genes in which SMA-9 represses expression, the *sma-9* loss-of-function phenotype is epistatic to that of *sma-3*, indicating that SMA-3 fails to regulate the expression of these genes in the absence of SMA-9.

## Biological functions of SMA-3 and SMA-9 target genes

Gene Ontology analysis showed that direct target genes of SMA-3 and SMA-9 shared some annotation clusters, including for fatty acid metabolism, collagens, and one-carbon metabolism (*Figure 3b*, *Figure 3—figure supplement 1*), but also showed that each regulated its own annotation clusters. SMA-3-exclusive direct targets were enriched for ribosome biogenesis factors, mitochondrial proteins, and ER chaperones (*Figure 3d*). Direct targets that were positively yet exclusively regulated by SMA-9 were enriched for genes involved in oxidation-reduction reactions and cytochrome P450s (*Figure 3e*), whereas direct targets that were negatively yet exclusively regulated by SMA-9 were enriched for innate immunity factors (*Figure 3f*).

Using a candidate gene approach, we previously identified several cuticle collagen genes that mediate the regulation of body size downstream of BMP signaling (*Madaan et al., 2018*). Here, we sought to validate and extend this analysis in an unbiased manner by screening target genes for a function in body size regulation. We selected 45 genes to analyze for a role in body size, selecting candidate genes so as to ensure a broad representation of the different types of regulation (e.g. co-regulated direct targets versus SMA-3 and SMA-9 exclusive direct targets), gene ontology (e.g. collagens, ER chaperones, lipid metabolism), and RNAi clone or mutant availability. For nine genes for which mutants were available, we measured body length at the L4 stage (*Figure 5*, *Figure 5—figure supplement 1*). We also measured *sma-3* and *sma-9* mutants as controls, finding them to be smaller, as expected. As we previously showed, *fat-6* and *fat-7* mutants were not significantly different from wild-type in body size (*Clark and Savage-Dunn, 2018b*). Of the remaining six genes for which mutants were available, only *ins-7* showed significantly reduced body size with a statistical effect size (Glass' effect size: the difference between the mean of the mutant and the mean for wild-type, divided by the standard deviation of wild-type) greater than one. To test the functions of genes for which mutants were not available, we used RNAi depletion in the *rrf-3* mutant (*Figure 5—figure supplement 1*). RRF-3 encodes an RNA-dependent RNA polymerase, and *rrf-3* mutants are often used in screens and phenotypic analyses because of their RNAi hypersensitivity (*Simmer et al., 2002*). RNAi depletion of controls *sma-3* and *sma-9* reduced the body length as expected. We chose 37 target genes to analyze, representing a variety of molecular functions and including SMA-3-exclusive and SMA-9-exclusive in addition to co-regulated genes (*Figure 5*).

Is there any correlation between whether a gene shows a large Glass effect size on body length and whether it is regulated by SMA-3 exclusively, SMA-9 exclusively, or co-regulated? We compared the reduction in body length caused by RNAi or mutation of each tested target (normalized as Glass' effect size in which larger values indicate smaller body lengths compared to control) against the adjusted p-value for body length compared to control for target genes in each of these three categories (*Figure 5a*). The genes that showed the largest and most statistically significant effect size were either SMA-3-exclusive or co-regulated. For each gene, we also compared body length effect size, gene ontology/classification, and the BETA rank scores for SMA-3 and SMA-9 ChIP-seq/RNA-seq data, which gives an indication of whether a gene is a direct target of one or both factors (*Figure 5b*). For each gene, we plotted the -log$_{10}$ for each BETA rank score (for SMA-3 on the X-axis and SMA-9 on the Y-axis) such that the greater the -log$_{10}$ BETA rank value, the greater probability that the gene is a direct target of that transcription factor. Circles represent individual genes tested such that the coordinate of the circle reflects the coordinate SMA-3 and SMA-9 BETA ranks, and the size of the circle represents the reduction in body size, normalized as Glass' effect size, relative to control. Circle color represents key associated GO terms for each gene. We found that the genes that promoted body length and belonged to the SMA-3-exclusive category encoded chaperones and collagen secretion factors. SMA-9-exclusive genes that promoted body length encoded a lectin and a gene involved in one-carbon metabolism. Co-regulated genes showing a role in body length also encoded one-carbon metabolism factors, as well as collagens and chaperones.

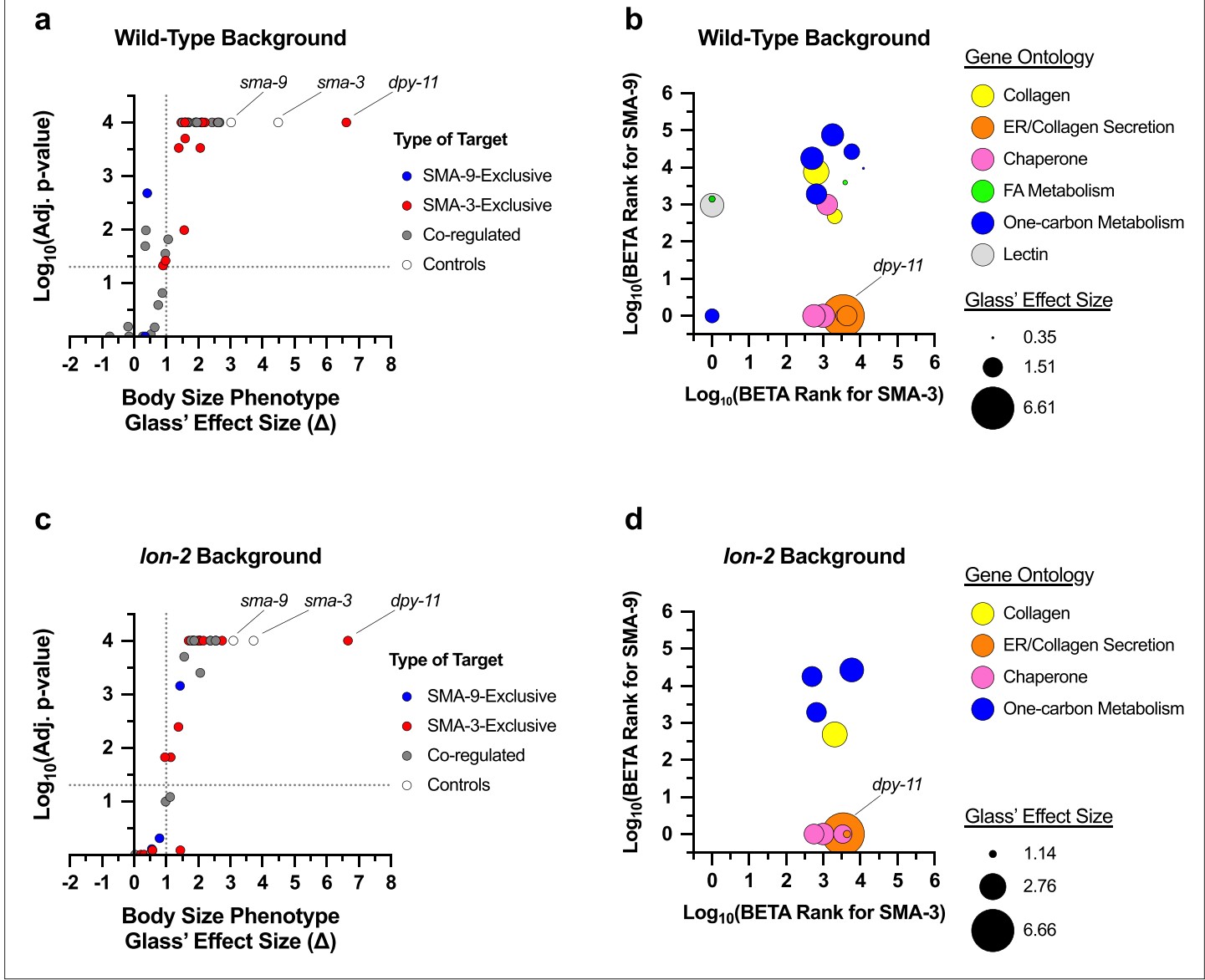

**Figure 5.** Multiple SMA-3 and SMA-9 target genes regulate body size. (**a,c**) The adjusted p-value (plotted as -log₁₀) for mean body size for genes either knocked down by RNAi or mutation compared to control (empty vector and wild-type, respectively) is shown for individual genes giving the indicated Glass' effect size Δ for body size when knocked down. Larger Glass' effect values indicate smaller bodies compared to the control. Genes regulated by SMA-9 exclusively, SMA-3 exclusively, or by both factors are indicated by blue, red, and gray circles, respectively. The horizontal dotted line indicates a p-value cutoff of 0.05. The vertical dotted line indicates an effect size cutoff of 1. The *sma-3* and *sma-9* controls are indicated by empty circles. (**b,d**) The BETA rank values for SMA-3 or SMA-9 (plotted as -log₁₀ and acting as measures of Chromatin immunoprecipitation sequencing (ChIP-seq)/RNA-seq correlation demonstrating the direct target nature of those factors) are shown as circles for individual genes. Each circle gives the Glass' effect size for body size (indicated by the area of each circle – larger circles indicate greater decreases in body size when the indicated gene is knocked down or mutated) and gene ontology group (indicated by circle color). Two genetic backgrounds are shown: (**a, b**) wild-type and (**c, d**) *lon-2*. The circle for *dpy-11* is highlighted.

The online version of this article includes the following figure supplement(s) for figure 5:

**Figure supplement 1.** Target genes required for exaggerated growth.

**Figure supplement 2.** Target genes required for exaggerated growth in *lon-2* mutants.

We reasoned that if target genes truly act downstream of the DBL-1 signaling pathway to regulate body size, then we would also expect them to act downstream of the negative regulator LON-2, a glycan which antagonizes the DBL-1 pathway with respect to body size at the level of ligand-receptor interactions (*Gumienny et al., 2007*). Thus, we also performed RNAi depletion in a *lon-2; rrf-3* double

mutant, which demonstrates exaggerated DBL-1 signaling and elongated body size; we expected that candidate transcriptional effectors of the DBL-1 pathway might have a more prominent requirement in mediating the exaggerated growth defect of a *lon-2* mutant and hence show a suppression phenotype in this genetic background. RNAi of one of the target genes, *mttr-1*, prevented the development of *rrf-3; lon-2* animals to the L4 stage, so body length could not be quantified at this stage. Nearly two-thirds of the remaining 36 genes tested caused a statistically significant reduction in body length (with a statistical effect size greater than one) upon RNAi treatment in at least one of the two genetic backgrounds. Half of the genes tested caused a significant reduction in body length in both genetic backgrounds (*Figure 5C*, *Figure 5—figure supplement 2*), making them strong candidates for direct transcriptional effectors of body size regulation. Consistent with our results in the wild-type background, we found that SMA-3-exclusive regulators of body size were enriched for chaperones and factors involved in ER secretion, whereas co-regulated genes mediating body size growth were enriched for one-carbon metabolism factors and collagens (*Figure 5d*), suggesting that the upregulation of these activities is a key aspect of how DBL-1 signaling promotes growth.

## DBL-1 signaling promotes body size through collagen secretion

We observed that *dpy-11* depletion via RNAi resulted in the most severe reduction in body length among the tested target genes (*Figure 5a and c*). The *dpy-11* gene encodes a protein-disulfide reductase involved in cuticle development (*Ko and Chow, 2003*), and the expression of this gene is regulated by SMA-3 but not by SMA-9 (*Figure 5b and d*). We hypothesized that *dpy-11* may represent a function for BMP signaling in cuticle collagen secretion, in addition to the previously established role in cuticle collagen gene expression. Furthermore, this role may be SMA-9-independent. We tested this hypothesis by monitoring the expression and localization of a cuticle collagen, ROL-6, a cuticle collagen gene with a demonstrated role in body size (*Madaan et al., 2018*). We previously showed that *rol-6* mRNA levels are reduced in *dbl-1* mutants at L2 (*Madaan et al., 2018*), although RNA-seq analysis did not find enough of a statistically significant change in *rol-6* to qualify it as a transcriptional target and total levels of protein are also not significantly reduced in mutants (*Figure 6f and g*).

The hypodermis synthesizes and secretes collagen into the cuticle in a pattern of circular annuli that surround the animal along its length (*Figure 6a*). These collagen-rich annuli and the newly synthesized collagen inside the hypodermal cells underneath the cuticle can be distinguished using confocal microscopy (*Figure 6b*). We generated a functional endogenously tagged allele of *rol-6* that expresses a ROL-6::wrmScarlet fusion protein, taking care to preserve the proteolytic processing sites such that wrmScarlet remains attached to the final protein product. As previously shown (*Aggad et al., 2023*), this fusion protein localizes to the cuticle (*Figure 6c*). In L4 animals at high magnification, the total cuticular fluorescence was reduced in both *sma-3* and *sma-9* mutants, with clear patches of decreased ROL-6::wrmScarlet (*Figure 6d and e*). Interestingly, the hypodermal subcellular distribution of ROL-6::wrmScarlet was altered in *sma-3* and *sma-9* mutants, with ROL-6::wrmScarlet protein accumulating intracellularly in these mutants (quantified in *Figure 6f*) compared to wild-type, often underneath the clear patches observed in the cuticular layer, which showed depressed levels of ROL-6::wrmScarlet protein in *sma-3* mutants (quantified in *Figure 6g*). This phenomenon is consistent with changes in collagen secretion upon impaired BMP signaling, and it suggests that this role is not SMA-9-independent. We used structured illumination super-resolution microscopy to compare an ER marker, VIT2ss::oxGFP::KDEL (*Tang et al., 2021*), with subcellular ROL-6::wrmScarlet, and we found that ROL-6 becomes trapped in and accumulates at the hypodermal ER in *sma-3* mutants (*Figure 7a–l*). We also noted that ER structures in *sma-3* mutants were thinner, with less complex and branched tubules, relative to wild-type (*Figure 7j and l*), consistent with a defect in secretion.

To assess the loss of the BMP signaling target *dpy-11* on collagen release, we depleted *dpy-11* via RNAi in nematodes expressing ROL-6::wrmScarlet. Consistent with a role for DPY-11 in the secretion of collagens, including ROL-6, we observed a severe disruption in both intracellular and cuticular deposition of ROL-6::wrmScarlet (*Figure 6h*; *Figure 6—figure supplement 1*). To validate that the perturbed ROL-6::wrmScarlet distribution and localization exhibited in *sma-3* and *sma-9* mutants (as well as *dpy-11* RNAi) was caused by disturbances in ER-specific processes, we targeted the ER chemically and genetically. We observed similar ROL-6::wrmScarlet subcellular distribution in tunicamycin-treated ROL-6::wrmScarlet expressing animals as seen in *sma-3* and *sma-9* mutants (*Figure 7m*; *Figure 7—figure supplement 1*). Tunicamycin inhibits the glycan biosynthesis pathway, disrupts ER-mediated

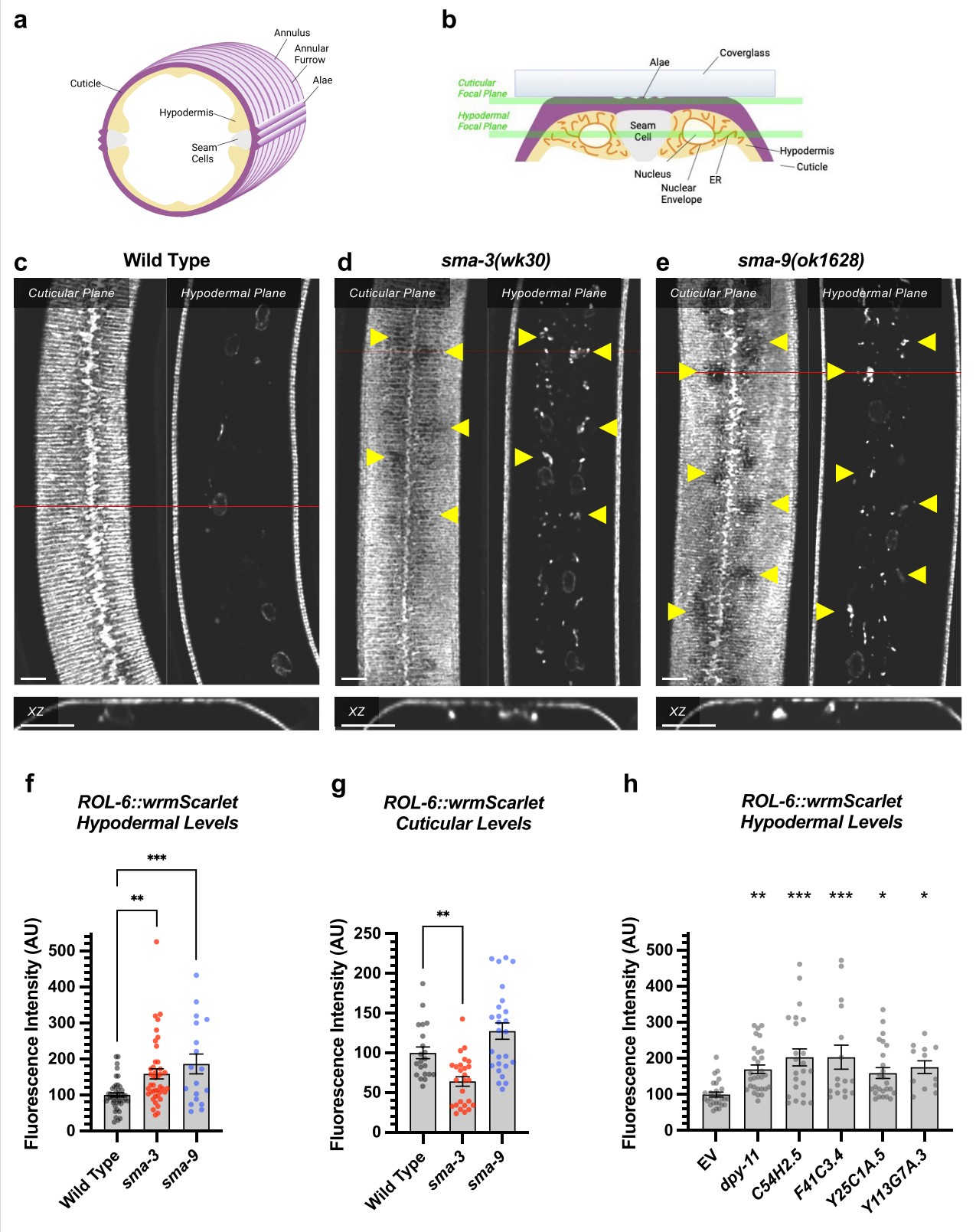

**Figure 6.** DBL-1 signaling promotes body size through collagen secretion. (**a**) Cartoon illustrating a cross-section through the nematode body (dorsal oriented up). Rings of cuticular collagen annuli (magenta), secreted by the underlying hypodermal cell layer (tan) surround the body. Lateral alae containing collagen, secreted by the underlying seam cells (gray), run orthogonal along the length of the body. (**b**) Cartoon illustrating a cross-section through a portion of the nematode body (lateral oriented up) underneath a microscope cover glass. Horizontal green bars indicate the cuticular versus

*Figure 6 continued on next page*

*Figure 6 continued*

hypodermal plates captured by confocal microscopy in panels c-e. (**c**) ROL-6::wrmScarlet fluorescence detected in annuli and alae in the cuticular layer, as well as in nuclear envelopes in the underlying hypodermal layer in wild-type animals. The horizontal red line indicates the specific xz cross section shown below the cuticular and hypodermal plane panels. The bar indicates 5 microns. (**d**) ROL-6::wrmScarlet in *sma-3* mutants, visualized as per wild-type. Patches of cuticular surface show diminished levels of ROL-6::wrmScarlet, whereas the protein is detected in the hypodermal layer just under these patches (yellow arrowheads), suggesting a failure to deliver collagen to the surface cuticle (easily visualized in the xz panel). (**e**) Mutants for *sma-9* show the same phenotype as *sma-3*. (**f,g**) Quantification of ROL-6::wrmScarlet fluorescence in the (**f**) hypodermal layer or (**g**) cuticular layer of indicated mutants. Dots indicate the fluorescence of individual animals. Asterisks over pairwise comparison bars indicate one-way ANOVA with (**f**) Sídák's multiple comparison test or (**g**) the Kruskall-Wallis comparison test (\*\*\*p<0.001, \*\*p<0.01). (**h**) A similar analysis for RNAi knockdowns of the SMA-3 target gene *dpy-11*, as well as four known ER secretion factors as comparative controls. Asterisks above each column indicate one-way ANOVA with Dunnett's multiple comparison test against wild-type (\*\*\*p<0.001, \*\*p<0.01, \*p<0.05). The bar indicates 5 microns.

The online version of this article includes the following figure supplement(s) for figure 6:

**Figure supplement 1.** DPY-11 promotes body size through collagen secretion.

secretion, and induces ER stress. Moreover, RNAi-mediated depletion of genes involved in various steps of ER-derived vesicle production and transport, including *C54H2.5* (SURF4 ortholog/ER cargo release), *F41C3.4* (GOLT1A/b ortholog/ER to Golgi apparatus transport), *Y25C1A.5* (COPB-1/COPI coat complex subunit), and *Y113G7A.3* (COPII coat complex subunit) increased ROL-6::wrmScarlet hypodermal intracellular accumulation and left empty patches in the cuticle, similar to BMP signaling mutants (*Figure 6g*; *Figure 6—figure supplement 1*). To test whether impaired secretion through the ER alone could compromise body size, we examined nematodes treated with tunicamycin and found that they showed a strong decrease in body size (*Figure 7n*). Taken together, our results suggest that signaling by DBL-1/BMP signaling promotes body size growth by promoting ER-specific processes involved in collagen maturation, transport, and secretion into the cuticular ECM.

## Discussion

### SMA-9/Schnurri is a Smad transcriptional partner with both joint and unique targets

TGF-β signaling pathways regulate sets of target genes to execute biological functions in a context-dependent manner. Canonical signaling is mediated by the Smad transcription factor complex, but Smad binding sites are too degenerate and have low affinity to account for the specific context-dependent effects, so transcription factor partners must also be involved. Some of the characterized partners are cell type-specific transcription factors. In contrast, Schnurri proteins are transcriptional partners that co-regulate target genes across multiple cell types. Here, we used genome-wide RNA-seq and ChIP-seq integrated through a novel software analysis pipeline to untangle the roles of Smad and Schnurri transcription factors in the developing *C. elegans* larva. We chose the second larval (L2) stage because Smad activity is elevated at this stage as determined by the RAD-SMAD activity reporter (*Tian et al., 2010*), and because this stage is the earliest point at which one can observe a clear difference for one of the best-studied Smad mutant phenotypes: body size growth.

Using ChIP-seq, we detected numerous SMA-3 and SMA-9 binding sites. The large number of sites could reflect the low affinity and degenerate DNA sequence recognition of known targets for their respective families of transcription factors. By using an analysis pipeline that combines BETA, which integrates ChIP-seq and RNA-seq data to identify targets, with LOA, which integrates two separate RNA-seq pairwise comparisons to identify shared DEGs, we identified nearby direct transcriptional targets based on their functional impact on transcript levels in corresponding mutants. Only a fraction of the total SMA-3 and SMA-9 sites were classified as functional based on BETA analysis of the ChIP-seq and RNA-seq datasets. Most SMA-3 and SMA-9 sites were not found at HOT sites, which are often considered to be non-specific binding sites typically found in open regions of chromatin. The high number of additional sites classified as non-functional could represent the detection of weak affinity targets that do not have an actual biological purpose. Alternatively, these sites could have an additional role in DBL-1 signaling besides transcriptional regulation of nearby genes, or they could be regulating the expression of target genes at a far enough distance to not be detected by our BETA analysis. The difference between total binding sites and those associated with changes in gene

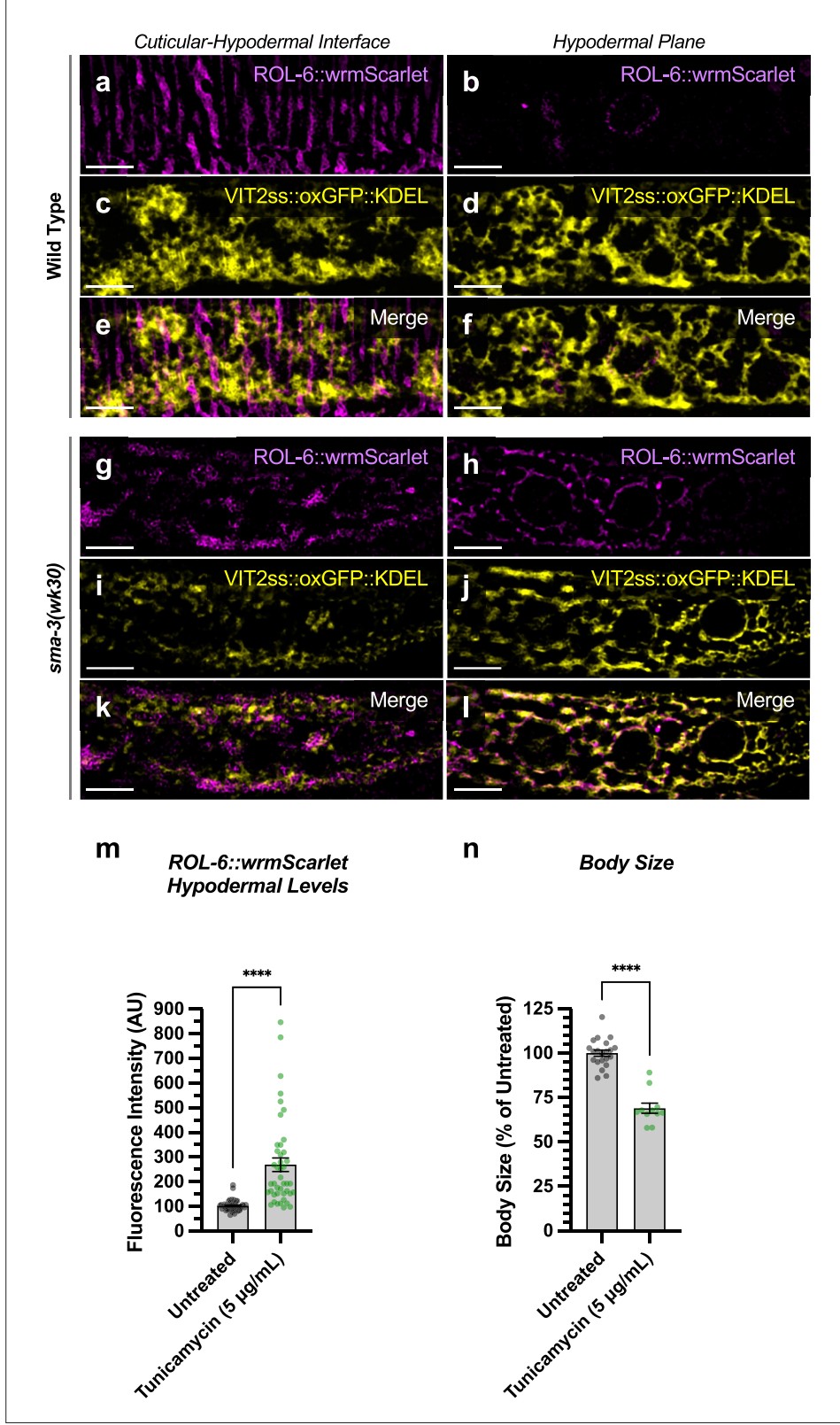

**Figure 7.** ROL-6::wrmScarlet accumulates in the endoplasmic reticulum (ER) of *sma-3* mutants. ROL-6::wrmScarlet (magenta) and VIT2ss::oxGFP::KDEL (an ER marker, shown here in yellow) shown separately (**a–d, g–j**) or merged (**e–f, k–l**) in either (**a–f**) wild-type or (**g–l**) a *sma-3* mutant. (**a,c,e,g,i,k**) shows the interface between the cuticle and the hypodermis, as visualized by SIM super-resolution microscopy. (**b, d, f, h, j, l**) shows the hypodermal

*Figure 7 continued on next page*

*Figure 7 continued*

layer at a focal plane centered around the nuclear envelope. In wild-type, most ROL-6::wrmScarlet is delivered into the cuticle. In *sma-3* mutants, lower levels of ROL-6::wrmScarlet are present in the cuticle and rapidly bleached under the SIM laser even under low power, whereas abundant ROL-6::wrmScarlet colocalized with the ER VIT2ss::oxGFP::KDEL marker (yellow). We noted that ER reticulation in *sma-3* was thinner and skeletonized compared to the wild-type, perhaps suggesting reduced secretory throughput. (**m**) Quantification of ROL-6::wrmScarlet fluorescence in the hypodermal layer of tunicamycin-treated versus untreated nematodes. (**n**) Mean body length of L4 animals (normalized to untreated) of tunicamycin treated versus untreated nematodes. (**m, n**) Dots indicate the values for individual animals. Asterisks over pairwise comparison bars indicate a student t-test (****$p<0.0001$). The bar indicates 5 microns.

The online version of this article includes the following figure supplement(s) for figure 7:

**Figure supplement 1.** Tunicamycin treatment mimics the effect of *sma-3* and *sma-9* mutations on collagen secretion.

---

expression underscores the importance of combining RNA-seq with ChIP-seq to identify the most biologically relevant targets.

## Functional interactions between SMA-3/Smad and SMA-9/Schnurri

Our analysis revealed that SMA-3/Smad and SMA-9/Schnurri have target genes that are co-regulated by both factors, as well as separate target genes that they independently and exclusively regulate (*Figure 8*). ChIP-seq data demonstrated that 73.7% of SMA-3 binding peaks overlap with SMA-9 binding sites, while approximately half of all SMA-9 binding sites did not overlap with SMA-3 sites. The significant number of shared (co-regulated) target genes with overlapping binding peaks is consistent with a model in which SMA-3 and SMA-9 bind as a complex (or at least adjacent along DNA), as has been demonstrated at a few loci in *Drosophila* and Xenopus. Our results extend this model to a genome-wide level. Further investigation will be needed to determine if SMA-3 and SMA-9 form a direct complex at these sites, and whether the presence of one factor affects the binding of the other.

Our combined BETA and LOA analysis further demonstrated that SMA-3/Smad acts primarily as a transcriptional activator, whereas SMA-9/Schnurri can function as either an activator or a repressor depending on the locus. These dual functions for SMA-9/Schnurri are consistent with previous studies that demonstrated that different domains of SMA-9 can act as activators or repressors in a heterologous system (*Liang et al., 2007*). Furthermore, SMA-9 DNA-binding domain fusions with known transcriptional activation or repression domains could each rescue a subset of mutant defects in *sma-9* mutants (*Liang et al., 2007*). Most co-regulated genes were activated by both SMA-3 and SMA-9, with a small subset activated by SMA-3 and repressed by SMA-9. We used double mutant analysis of *sma-3; sma-9* animals to determine how these factors interact to produce normal body size. While both single mutants were phenotypically small, the *sma-9* mutation partially suppressed the small body size of *sma-3* mutants so that double mutants were intermediate in size, suggesting that these factors may have antagonistic interactions in addition to the expected cooperative effects. Antagonism would be consistent with the observed interaction between SMA-9 and the BMP pathway in mesodermal lineage specification (*Foehr et al., 2006*; *Liu et al., 2015*), and with the observation of factors that have both positive and negative effects on body size (*DeGroot et al., 2023*).

We used RT-PCR to determine how SMA-3 and SMA-9 function together at a locus-specific level. For co-activated target genes, genetic analysis was consistent with the proteins acting together rather than additively. In the *sma-3; sma-9* double mutant, genes regulated in opposite directions by SMA-3 and SMA-9 showed upregulation as in *sma-9* single mutants. Thus, SMA-3 activation of these genes is dependent on the presence of SMA-9. This dependence could occur if SMA-9 is needed to recruit SMA-3 to the DNA. This possibility would be a novel mode of interaction, because, for the previously analyzed *brinker* and *Xvent2* target genes, Schnurri was shown to be recruited by the Smad complex rather than vice versa. A second possibility is that SMA-3 can bind in the absence of SMA-9 but cannot engage with the transcriptional machinery; that is, both proteins are required to form a transcriptional activation complex. With either possibility, SMA-9 represses the expression of these genes in the absence of SMA-3.

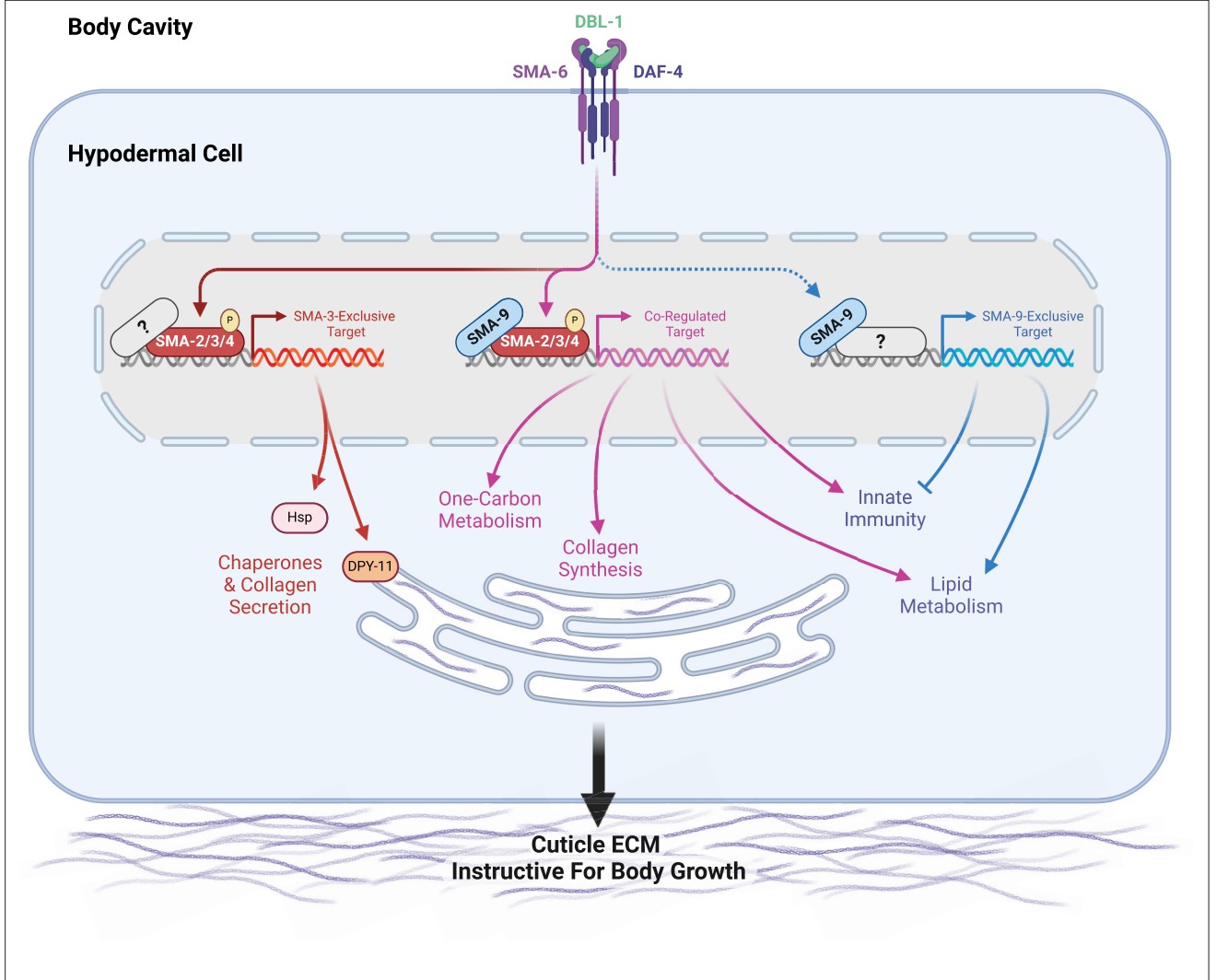

**Figure 8.** A model for the regulation of growth by DBL-1/BMP Signaling. During early larval development, the DBL-1 ligand binds to the BMP receptors SMA-6 and DAF-4, which activate the Smads SMA-2, SMA-3, and SMA-4. The resulting Smad complex binds to one category of sites along the genome either alongside or in complex with SMA-9, co-regulating neighboring genes (in purple). These co-regulated genes include several collagen genes, factors involved in one-carbon metabolism, innate immunity genes, and genes involved in lipid metabolism. The Smad complex also binds to another category of sites (in orange/red) which lack SMA-9, perhaps associating instead with other transcription factors or co-factors (gray question mark). These SMA-3-exclusive genes include chaperones and the disulfide reductase DPY-11, which in turn promote the secretion of collagen into the cuticular extracellular matrix, thereby remodeling the cuticle to allow for growth. In addition to binding either with or near Smad complex components, SMA-9 (in blue) also binds to sites along the genome lacking Smad (or at least SMA-3), perhaps associating instead with other transcription factors or co-factors (gray question mark). These SMA-9-exclusive genes, which can be either positively or negatively regulated by SMA-9, play a minimal role in body size growth, but rather are associated with innate immunity and lipid metabolism.

## Biological functions of target genes

DBL-1 signaling regulates multiple developmental and physiological processes in *C. elegans*, including body size, lipid metabolism, innate immunity, and male tail development. Our samples were not enriched for males, precluding an analysis of targets involved in male tail development. Binding sites and their nearby regulated targets fell into three classes: SMA-3-exclusive, SMA-9-exclusive, and co-regulated. Interestingly, the GO terms for both shared and independent target genes partially overlap, suggesting broad similarity in biological functions. Within the subset of SMA-3-exclusive target genes, we noticed GO term enrichment for chaperones and factors involved in collagen secretion. Given the low affinity and degenerate nature of Smad binding sites, we speculate that additional

binding factors associate with SMA-3 at these SMA-3-exclusive targets to facilitate regulation. Future studies will be needed to identify these novel partners.

By contrast, the subset of SMA-9-exclusive target genes was enriched in GO terms associated with lipid metabolism and innate immunity. Although SMA-3 functions in both fat storage and pathogen resistance, respectively (*Clark et al., 2018a*; *Mallo et al., 2002*; *Yu et al., 2017*; *Clark et al., 2021*; *Zugasti and Ewbank, 2009*; *Ciccarelli et al., 2024a*; *Ciccarelli et al., 2024b*), these SMA-9-exclusive target genes imply Smad-independent roles for SMA-9 in these functions. A Smad-independent role for SMA-9 in immunity is consistent with the pronounced role of vertebrate Schnurri homologs in immunity (*Jones and Glimcher, 2010*), which have not been reported to overlap with TGF-β-regulated functions. In vertebrates, Schnurri homologs are shown to be direct DNA-binding proteins with diverse biological functions that include TGF-β-responsive and TGF-β-independent roles. In TGF-β-independent roles, they bind NFκB-like sequences (*Gaynor et al., 1991*), and can interact with other transcription factors including TRAF2 and c-Jun (*Oukka et al., 2002*; *Oukka et al., 2004*). It will be interesting to determine whether the SMA-9/Schnurri-exclusive target genes are responsive to TGF-β signals and/or to other signaling ligands.

## Identification of target genes that mediate body size regulation

GO term analysis readily identified target genes involved in lipid metabolism and pathogen response, but target genes required for body size regulation remain more difficult to predict based on sequence alone. Furthermore, we have previously shown that the body size and lipid metabolism functions are separable (*Clark et al., 2018a*). We, therefore, conducted a functional analysis of these target genes by performing body size measurements on their corresponding mutants. We also performed body size measurements on RNAi knockdowns for identified target genes in an RNAi-sensitive strain, examining their effect in both a wild-type background and a *lon-2* background in which DBL-1 signaling is exaggerated, resulting in an elongated body size. Normalizing these data using Glass' effect size allowed us to make broad comparisons between mutants and RNAi knockdowns. These analyses confirmed previous work focusing on the role of the cuticle in mediating body size regulation by DBL-1/BMP (*Madaan et al., 2018*). Although RNAi knockdowns of *clec-1, fah-1, C52D10.3, dre-1, hsp-12.3, wrt-1* reduced body size in the *rrf-3* mutant background, they failed to reduce the body size in *lon-2; rrf-3* mutants, suggesting that they regulate body size upstream or independently of the DBL-1 pathway. By contrast, RNAi knockdowns of *haf-9, his-32, zip-10, emb-8, F25B5.6, nath-10*, and *hsp-3* reduced body size in *lon-2; rrf-3* double mutants but not in *rrf-3* single mutants, suggesting that they might be factors whose effects are only detectable in the context of an overactive pathway; these warrant future study. Seventeen genes showed a body reduction with a statistically significant effect size equal to or greater than one in both genetic backgrounds, making them of particular interest. These genes had GO terms associated with either one-carbon metabolism or chaperone/ER secretion, suggesting that the upregulation of these activities is a key aspect of how DBL-1 signaling promotes growth (*Figure 8*).

How could one-carbon metabolism play a role in body size growth? This complex set of interlinked metabolic cycles is critical for methionine and folate homeostasis. It also provides the methyl groups needed to synthesize nucleotides, amino acids, the antioxidant glutathione, creatine, and phospholipids like phosphatidylcholine, a fundamental component of membranes (*Clare et al., 2019*). One-carbon metabolism also provides the methyl groups needed to make epigenetic marks on DNA and chromatin. The specific role of this metabolic pathway in body size growth will be an important topic of future study.

As in previous datasets, this RNA-seq analysis identified collagen genes (*col-94* and *col-153*) as direct co-regulated targets of SMA-3 and SMA-9. An enrichment of ER secretion and chaperone factors in the list of direct targets involved in body size growth was unexpected, but reasonable given that one of the key functions of the hypodermis is to secrete cuticular collagen. The role of collagens in body size and morphology is well documented (*Page and Johnstone, 2007*; *McMahon et al., 2003*), and the secreted ADAMTS metalloprotease ADT-2 modifies cuticle collagen organization and regulates body size (*Fernando et al., 2011*). The thioredoxin-like DPY-11 was a particularly compelling target given its established role in cuticle formation and the dramatic effect of its loss on body size growth, as well as the known role of these enzymes in processing secreted proteins moving through the ER/Golgi network (*Ko and Chow, 2003*). To test whether DBL-1 regulates ER secretion of

collagen, we turned to endogenously tagged cuticle collagen ROL-6::wrmScarlet to analyze subcellular localization. Using this reporter for collagen synthesis and secretion, we demonstrated that the DBL-1 pathway influences the secretion of this cuticle collagen. In particular, ROL-6::wrmScarlet accumulates in a perinuclear ER compartment in both *sma-3* and *sma-9* mutants. Consistent with a secretion defect, we found a corresponding decrease in the amount of ROL-6::wrmScarlet in the cuticle of *sma-3* mutants, although not in *sma-9* mutants, which could reflect the differential enrichment of *dpy-11* and chaperones in the list of SMA-3-exclusive target genes. Interestingly, treatment with tunicamycin, which impairs ER secretion, was sufficient to reduce body size, which is consistent with a model in which BMP signaling promotes collagen secretion to foster growth (*Figure 8*). We remain cautious in our interpretation of this result, as blocking ER secretion with tunicamycin could affect the secretion of the BMP receptors or other proteins that function together with the receptors, which could also lead to a body size defect.

The collagenous cuticle is a major target of DBL-1/BMP signaling in body size regulation. For example, others have also demonstrated that DBL-1 signaling regulates cuticle collagen LON-3 post-transcriptionally (*Suzuki et al., 2002*). In addition to direct transcriptional regulation of cuticle components, here, we highlight the importance of regulating other target genes such as *dpy-11* that are needed for collagen post-transcriptional processing. One powerful element of our approach was that our ChIP-seq/RNA-seq BETA analysis identified direct target genes and combined those findings with a functional analysis of a subset of those targets, thereby demonstrating that a combination of SMA-3-exclusive targets (including chaperones and collagen secretion factors) work together with SMA-3/SMA-9 co-regulated targets (including collagen genes and factors involved in one-carbon metabolism) to affect the process of body growth through their regulation of the extracellular matrix of the surrounding cuticle. TGF-β family members are well-known regulators of collagen deposition and extracellular matrix composition, suggesting that this class of transcriptional targets is conserved over evolutionary time (*Kim et al., 2018*; *MacFarlane et al., 2017*). Thus, it is likely that the multi-level interactions identified in *C. elegans* are also relevant to the functions of these factors in vertebrates.

# Materials and methods

**Key resources table**

| Reagent type (species) or resource | Designation | Source or reference | Identifiers | Additional information |
|---|---|---|---|---|
| Genetic reagent (*C. elegans*) | N2 | CGC | | |
| Genetic reagent (*C. elegans*) | CS24 | Savage-Dunn lab | *sma-3(wk30)* | |
| Genetic reagent (*C. elegans*) | VC1183 | CGC | *sma-9(ok1628)* | |
| Genetic reagent (*C. elegans*) | GFP::SMA-3 | Savage-Dunn lab | *qcIs6[sma-3p::gfp::sma-3, rol-6(d)]* | |
| Genetic reagent (*C. elegans*) | SMA-9::GFP | Liu lab | *jjIs1253[sma-9p::sma-9C2::gfp +unc-119(+)]* | |
| Genetic reagent (*C. elegans*) | ROL-6::wrmScarlet | Savage-Dunn lab | *rol-6(syb2235[rol-6::wrmScarlet])* | |
| Genetic reagent (*C. elegans*) | ER marker | Barth Grant | *pwSi82[hyp-7p::VIT2ss::oxGFP::KDEL, HygR].* | |
| Genetic reagent (*C. elegans*) | NL2099 | CGC | *rrf-3(pk1426)* | |
| Genetic reagent (*C. elegans*) | BX106 | CGC | *fat-6(tm331)* | |
| Genetic reagent (*C. elegans*) | BX153 | CGC | *fat-7(wa36)* | |
| Genetic reagent (*C. elegans*) | VC1760 | CGC | *nhr-114(gk849)* | |
| Genetic reagent (*C. elegans*) | CB7468 | CGC | *acs-22(gk373989)* | |
| Genetic reagent (*C. elegans*) | VC4077 | CGC | *lbp-8(gk5151)* | |
| Genetic reagent (*C. elegans*) | CB6734 | CGC | *clec-60(tm2319)* | |
| Genetic reagent (*C. elegans*) | VC2477 | CGC | *sysm-1(ok3236)* | |
| Genetic reagent (*C. elegans*) | RB1388 | CGC | *ins-7(ok1573)* | |
| Genetic reagent (*C. elegans*) | CB502 | CGC | *sma-2(e502)* | |

*Continued on next page*

*Continued*

| Reagent type (species) or resource | Designation | Source or reference | Identifiers | Additional information |
|---|---|---|---|---|
| Sequence-based reagent | *act-1f* | This paper | 5'-ATGTGTGACGACGAGGTTGCC-3' | qRT-PCR primer |
| Sequence-based reagent | *act-1r* | This paper | 5'-GTCTCCGACGTACGAGTCCTT-3' | qRT-PCR primer |
| Sequence-based reagent | *fat-6f* | This paper | 5'-GTGGATTCTTCTTCGCTCAT-3' | qRT-PCR primer |
| Sequence-based reagent | *fat-6r* | This paper | 5'-CACAAGATGACAAGTGGGAA-3' | qRT-PCR primer |
| Sequence-based reagent | *nhr-114f* | This paper | 5'-CATTCGATGTTTTTGAGGCG-3' | qRT-PCR primer |
| Sequence-based reagent | *nhr-114r* | This paper | 5'-GATCGAAGTAGGCACCATCT-3' | qRT-PCR primer |
| Sequence-based reagent | *C54E4.5f* | This paper | 5'-GGCAGGTCTAATCCACGACTTG-3' | qRT-PCR primer |
| Sequence-based reagent | *C54E4.5r* | This paper | 5'-CTAATGTCCGGGTTCCCATCG-3' | qRT-PCR primer |
| Sequence-based reagent | *aard-19f* | This paper | 5'-CGGAGGTTACGAGACCAGTACG-3' | qRT-PCR primer |
| Sequence-based reagent | *aard-19r* | This paper | 5'-TGGAGTCACAGACGGAAGACG-3' | qRT-PCR primer |
| Sequence-based reagent | *nspe-7f* | This paper | 5'-CTCCAAACCTTCTTTTCTCCTTCG-3' | qRT-PCR primer |
| Sequence-based reagent | *nspe-7r* | This paper | 5'-GGACCGCCAGCCATATTGTC-3' | qRT-PCR primer |
| Sequence-based reagent | *nspc-16f* | This paper | 5'-TGTTCTCCATGGTTGAGTTATGCT-3' | qRT-PCR primer |
| Sequence-based reagent | *nspc-16r* | This paper | 5'-GTTTCTTTGCGGGGAATGTTGC-3' | qRT-PCR primer |
| Sequence-based reagent | *catp-3f* | This paper | 5'-TTCGGTTGGAGGTGTCGTTG-3' | qRT-PCR primer |
| Sequence-based reagent | *catp-3r* | This paper | 5'-GTTGCTCGGCATTCAGTACG-3' | qRT-PCR primer |
| Sequence-based reagent | *gdh-1f* | This paper | 5'-TGCTCGTGGAGATTGCCTCATC-3' | qRT-PCR primer |
| Sequence-based reagent | *gdh-1r* | This paper | 5'-GCATCTTGTTGGCTTCCTCGTC-3' | qRT-PCR primer |

## *C. elegans* strains

*C. elegans* strains were grown at 20 °C using standard methods unless otherwise indicated. N2 is the wild-type strain; some strains were provided by the *Caenorhabditis Genetics Center* (CGC), which is funded by the NIH Office of Research Infrastructure Programs (P40 OD010440) or generated in previous work. Strong loss-of-function or null alleles were used. The following genotypes were used: *sma-3(wk30)*, *sma-9(ok1628)*, *qcIs6[sma-3p::gfp::sma-3, rol-6(d)]*, *jjIs1253[sma-9p::sma-9C2::gfp +unc-119(+)]* (generated via bombardment), (**Praitis et al., 2001**), *rrf-3(pk1426)*, *fat-6(tm331)*, *fat-7(wa36)*, *nhr-114(gk849)*, *acs-22(gk373989)*, *lbp-8(gk5151)*, *clec-60(tm2319)*, *sysm-1(ok3236)*, *ins-7(ok1573)*, *sma-2(e502)*, *rol-6(syb2235[rol-6::wrmScarlet])* (**Aggad et al., 2023**), and *pwSi82[hyp-7p::VIT2ss::oxGFP::KDEL, HygR]*. The double mutants *sma-3(wk30); sma-9(ok1628)*, and *rrf-3(pk1426); lon-2(e678)* were generated in this study.

## RNA-seq

Developmentally synchronized animals were obtained by hypochlorite treatment of gravid adults to isolate embryos. Animals were grown on NGM plates at 20 °C until the late L2 stage. Total RNA was isolated from animals using Trizol (Invitrogen) combined with Bead Beater lysis in three biological replicates for each genotype (**Vora et al., 2022**). Libraries were generated using polyA selection in a paired-end fashion and sequenced on an Illumina HiSeq (2×150 bp configuration, single index, per lane) by Azenta (formerly Genewiz). Reads were mapped to the *C. elegans* genome (WS245) and gene counts were generated with STAR 2.5.1 a. Normalization and statistical analysis on gene counts were performed with EdgeR using generalized linear model functionality and tagwise dispersion estimates. Principal Component Analysis showed tight clustering within four biological replicates, with a clear separation between SMA-3 and SMA-9 active versus inactive genotypes. Likelihood ratio tests were conducted in a pairwise fashion between genotypes with a Benjamini and Hochberg correction. All RNA-seq raw sequence files as well as normalized counts after EdgeR can be accessed at GEO (Accession Number: GSE266398).

## Chromatin immunoprecipitation sequencing

Chromatin immunoprecipitation sequencing (ChIP-seq) using a poly-clonal goat IgG anti-GFP antibody (a gift from Tony Hyman and Kevin White) was performed on L2 stage nematodes by Michelle Kudron (Valerie Reinke Model Organism ENCyclopedia Of DNA Elements and model organism Encyclopedia of Regulatory Networks group) on the *sma-3(wk30);qcIs6[GFP::SMA-3]* and LW1253*: jjIs1253[sma-9p::sma-9C2::gfp +unc-119(+)]* strains at the late L2 stage (*Gerstein et al., 2010*); Data are available at https://www.encodeproject.org/. To calculate distances between SMA-3 and SMA-9 ChIP-seq peaks, each peak was reduced to a centroid position (midpoint between the two border coordinates along the chromosome). For each chromosome, a matrix of SMA-3 and SMA-9 peak centroids was created, allowing the measurement of distance (in bps) between every SMA-3 and SMA-9 centroid along that chromosome. The shortest distance in the matrix was chosen to define each SMA-3/SMA-9 nearest neighboring pair. The resulting inter-centroid distances were analyzed from all six chromosomes. To mimic a random distribution of SMA-3 and SMA-9 peaks, each peak on a given chromosome was reassigned to a location on that chromosome using randomized values (generated by the Microsoft Excel randomization function) within the size range for that chromosome. A matrix of SMA-3 and SMA-9 peak centroids was then analyzed from this randomized dataset as described above for the actual dataset.

## Identification of direct targets using BETA and LOA

To identify SMA-3 direct targets, differentially expressed genes (DEGs) from the RNA-seq comparison of wild-type versus *sma-3(wk30)* using an FDR ≤0.05 were compared against the genomic coordinates of SMA-3 peaks from the ChIP-seq analysis using BETA basic and the WS245 annotation of the *C. elegans* genome (*Wang et al., 2013*). The following parameters were used: 3 kb from TSS, FDR cutoff of 0.05, and one-tail KS test cutoff of 0.05. The input files consisted of.bed files of IDR thresholded peaks and differential expression Log$_2$FC and FDR values from the RNA-seq. An identical approach was used to identify SMA-9 direct targets using DEGs from the RNA-seq comparison of wild-type versus *sma-9(ok1628)*.

To identify direct targets co-regulated by both SMA-3 and SMA-9, the two pairwise RNA-seq comparisons (wild-type versus *sma-3* and wild-type versus *sma-9*) were analyzed, measuring DEGs for the same genes in both comparisons. Taking a conditional approach, the information from the first comparison (wild-type versus *sma-3*) was examined to see if it affected interpretation in the second (wild-type versus *sma-9*). Using the approach of *Luperchio et al., 2021*, the genes in the second comparison were split into two groups, conditional on the results in the first comparison, with one group comprising genes found to show differential expression in the first comparison, and the second group comprising genes found not to show differential expression. To estimate which genes were differentially regulated, an FDR of 0.01 was used to generate an overlapping list between the two comparisons. BETA basic was then used to identify potential direct targets of the SMA-3/SMA-9 combination using just the ChIP-seq peaks that overlapped between the two transcription factors. The following parameters were used: 3 kb from TSS, FDR cutoff of 0.05 and one-tail KS test cutoff of 0.05. Analysis tools can be obtained at GitHub: https://github.com/shahlab/hypoxia-multiomics (*Shah, 2022*) as per *Vora et al., 2022*. The WormBase database was used to obtain information about candidate target genes, including sequence, genetic map position, expression pattern, and available mutant alleles (*Davis et al., 2022*).

## Quantitative RT-PCR analysis

Worms were synchronized using overnight egg lay followed by 4 hr synchronization. When animals reached the L2 stage, they were collected and washed, and then RNA was extracted using a previously published protocol *Yin et al., 2015* followed by Qiagen RNeasy miniprep kit (Catalogue. No. 74104). Invitrogen SuperScript IV VILO Master Mix (Catalogue. No.11756050) was used to generate cDNA, and qRT-PCR analysis was done using Applied Biosystems *Power* SYBR Green PCR Master Mix (Catalogue. No. 4367659). Delta delta Ct analysis was done using Applied Biosystems and StepOne software. All qRT-PCR analysis was repeated on separate biological replicates. The following primer pairs were used: 5'-ATGTGTGACGACGAGGTTGCC-3' and 5'-GTCTCCGACGTACGAGTCCTT-3' to detect *act-1*, 5'-GTGGATTCTTCTTCGCTCAT-3' and 5'-CACAAGATGACAAGTGGGAA-3' to detect *fat-6*, 5'-CATTCGATGTTTTTGAGGCG-3' and 5'-GATCGAAGTAGGCACCATCT-3' to detect *nhr-114*,

5'-GGCAGGTCTAATCCACGACTTG-3' and 5'-CTAATGTCCGGGTTCCCATCG-3' to detect *C54E4.5*, 5'-CGGAGGTTACGAGACCAGTACG-3' and 5'-TGGAGTCACAGACGGAAGACG-3' to detect *aard-19*, 5'-CTCCAAACCTTCTTTTCTCCTTCG-3' and 5'-GGACCGCCAGCCATATTGTC-3' to detect *nspe-7*, 5'-TGTTCTCCATGGTTGAGTTATGCT-3' and 5'-GTTTCTTTGCGGGGAATGTTGC-3' to detect *nspc-16*, 5'-TTCGGTTGGAGGTGTCGTTG-3' and 5'-GTTGCTCGGCATTCAGTACG-3' to detect *catp-3*, and 5'-TGCTCGTGGAGATTGCCTCATC-3' and 5'-GCATCTTGTTGGCTTCCTCGTC-3' to detect *gdh-1*. All graphs were made using GraphPad Prism software and statistical analysis was performed using One-way ANOVA with Multiple Comparison Test, as calculated using the GraphPad software. There were two biologically independent collections from which three cDNA syntheses were analyzed using two technical replicates per data point.

## RNAi analysis of body size

RNAi knockdown of individual target genes was performed in the RNAi-sensitive *C. elegans* mutants *rrf-3(pk1426)* and *rrf-3(pk1426); lon-2(e678),* which were fed HT115 bacteria containing dsRNA expression plasmid L4440, with or without gene targeting sequences between flanking T7 promoters. NGM growth plates were used containing ampicillin for L4440 RNAi plasmid selection and IPTG (isopropyl β-D-1-thiogalactopyranoside) to induce dsRNA expression. Both *rrf-3(pk1426)* and *rrf-3(pk1426); lon-2(e678)* were exposed to the RNAi food during the L4 stage and allowed to lay eggs for 3 hr and then removed. Following hatching and development to adulthood while exposed to the RNAi food, 2 adult hermaphrodites were transferred to fresh RNAi plates, allowed to lay eggs, and removed from the plate. Upon hatching and development to the L4 stage, hermaphrodites were imaged using an Axio-Imager M1m (Carl Zeiss, Thornwood, NY) with a 5 X (NA 0.15) objective. The RNAi feeding constructs were obtained from the Open Biosystems library (Invitrogen), except for *C54E4.5* RNAi, which was constructed in this study. To analyze the body length of the RNAi-exposed animals, three independent measurements were made per worm using the segmented line tool on Fiji/ImageJ (*Schindelin et al., 2012*). Three to five biological replicates were completed for each RNAi construct. The data were analyzed using ANOVA with Dunnett's post hoc test correction for multiple comparisons.

## Hypodermal imaging of ROL-6::wrmScarlet

Animals expressing ROL-6::wrmScarlet in different genetic backgrounds were imaged using a Chroma/89 North CrestOptics X-Light V2 spinning disk, a Chroma/89North Laser Diode Illuminator, and a Photometrics PRIME95BRM16C CMOS camera via MetaMorph software. Day 1 adults (unless otherwise noted) were used to ensure molting was completed. A 63 X oil objective (NA 1.4) was used to detect fluorescence. In order to visualize the cuticle and hypodermis layers of each animal, a z-series was completed using a 0.5- micron step size across 6.5 microns. Each image was analyzed using Fiji/ImageJ for fluorescence quantification in the hypodermis of the animals. Background was subtracted using a rolling ball filter. An outline was drawn around each nematode and the mean fluorescence intensity was calculated within the outline. At least 10 animals were analyzed and pooled across three to four biological replicates. Using GraphPad Prism, the individual mean fluorescence intensity values were normalized to the mean for control animals in each experiment and analyzed using ANOVA with Dunnett's post hoc test correction for multiple comparisons. Images were then deconvolved using DeconvolutionLab2 (*Sage et al., 2017*).

Animals expressing ROL-6::wrmScarlet together with the ER marker VIT2ss::oxGFP::KDEL were imaged using a Zeiss Elyra 7 Lattice SIM. A 60 X water objective (NA 1.2) was used to detect fluorescence, and a z-series was completed as described above.

RNAi treatment of ROL-6::wrmScarlet animals was performed similarly to the RNAi treatment in the body length analysis with these exceptions: nematodes exposed to *dpy-11* RNAi grew for one generation until day 1 adulthood before imaging, nematodes exposed to C54H2.5 and F41C3.4-containing RNAi plasmids were introduced to the animals at the L1 development stage and allowed to develop until day 1 adulthood, and nematodes exposed to Y25C1A.5 and Y113G7A.3-containing RNAi plasmids were introduced to animals at the L4 development stage and grown for 24 hr before imaging. Tunicamycin treatment of ROL-6::wrmScarlet-expressing animals was completed by allowing animals to the develop from eggs to L4 stage in the presence of 5 µg/mL in NGM plates. Experiments were conducted over three to four biological replicates. The data were analyzed using an unpaired

two-tailed t-test ANOVA with Dunnett's post hoc test correction for multiple comparisons, where appropriate.

## Acknowledgements

We thank Michelle Kudron with the Model Organism ENCyclopedia of DNA Elements and model organism Encyclopedia of Regulatory Networks projects for performing chromatin immunoprecipitation sequencing. Some strains were provided by the *Caenorhabditis* Genetics Center, which is funded by the National Institutes of Health (NIH) Office of Research Infrastructure Programs (P40 OD-101440). We thank Barth Grant for the *pwSi82* strain, and Nanci Kane for her assistance with the Zeiss Elyra 7 Lattice SIM within the Waksman Institute Shared Imaging Facility (Rutgers, The State University of New Jersey). We thank Derek Gordon for his guidance and assistance with the analysis of the ChIP-seq distance matrix. This work was supported by NIH grants R15GM112147 to CSD, R21AG075315 to CSD and CR, R35GM130351 to JL, and R01GM101972 to CR.

## Additional information

### Competing interests

Mehul Vora: Works at ModOmics Ltd, no other competing interests to declare. The other authors declare that no competing interests exist.

### Funding

| Funder | Grant reference number | Author |
|---|---|---|
| National Institutes of Health | R15GM112147 | Cathy Savage-Dunn |
| National Institutes of Health | R21AG075315 | Christopher Rongo Cathy Savage-Dunn |
| National Institutes of Health | R35GM130351 | Jun Kelly Liu |
| National Institutes of Health | R01GM101972 | Christopher Rongo |

The funders had no role in study design, data collection and interpretation, or the decision to submit the work for publication.

### Author contributions

Mehul Vora, Investigation, Methodology, Writing – review and editing; Jonathan Dietz, Zachary Wing, Investigation, Writing – review and editing; Karen George, Investigation; Jun Kelly Liu, Resources, Funding acquisition, Writing – review and editing; Christopher Rongo, Conceptualization, Supervision, Funding acquisition, Writing – original draft; Cathy Savage-Dunn, Conceptualization, Resources, Supervision, Investigation, Writing – original draft, Project administration

### Author ORCIDs

Christopher Rongo  https://orcid.org/0000-0002-1361-5288
Cathy Savage-Dunn  https://orcid.org/0000-0002-3457-0509

Reviewer #1 (Public review): https://doi.org/10.7554/eLife.99394.3.sa1
Author response https://doi.org/10.7554/eLife.99394.3.sa2

## Additional files

### Supplementary files

Supplementary file 1. SMA-3 and SMA-9 Chromatin immunoprecipitation sequencing (ChIP-seq) sites. This file contains the chromosomal location of 4205 ChIP-seq peaks for SMA-3 and 7065 ChIP-

seq peaks for SMA-9 in separate tabs labeled 'SMA-3' and 'SMA-9,' respectively. SMA-3 sites that overlap with a SMA-9 site are listed on the 'Overlapping Sites_S3' tab. SMA-9 sites that overlap with a SMA-3 site are listed on the 'Overlapping Sites_S9' tab. Non-overlapping SMA-3 and SMA-9 sites are listed on the 'Non-overlapping_S3' and 'Non-overlapping_S9' tabs, respectively. For all tabs, column A indicates chromosome location, column B indicates the start of the peak sequence, and column C indicates the end of the peak sequence. Column labels are in row 1.

Supplementary file 2. Differential gene expression from *sma-3* and *sma-9* mutants. This file lists the differential gene expression from RNA-seq of *sma-3* versus wild-type (the tab labeled 'SMA3 versus N2'), as well as *sma-9* versus wild-type (the tab labeled 'SMA9 versus N2'). For each gene, WormBase GeneID, public gene name, log fold change, p-value, and false discovery rate (FDR) are listed. Column labels are in row 1.

Supplementary file 3. SMA-3 and SMA-9 direct targets. This file lists the direct target genes identified by BETA analysis of RNA-seq and Chromatin immunoprecipitation sequencing (ChIP-seq) data. Direct targets of SMA-3 are in the tab labeled 'SMA3 Direct Targets.' Direct targets of SMA-9 are in the tab labeled 'SMA9 Direct Targets.' For each gene, chromosomal location, transcriptional start site, transcriptional end site, public gene name, rank product from BETA analysis, RNA-seq log fold change from corresponding mutant versus wild-type, and WormBase GeneID are listed. Column labels are in row 1.

Supplementary file 4. Differential gene expression shared between *sma-3* and *sma-9* mutants analyzed using LOA. This file lists the differentially expressed genes identified by LOA analysis as being common to both the RNA-seq of *sma-3* versus wild-type as well as the RNA-seq of *sma-9* versus wild-type. For each gene, WormBase GeneID and public gene name are listed, followed by the log fold change, p-value, and false discovery rate (FDR) from the *sma-3* versus wild-type RNA-seq, followed by the log fold change, p-value, and FDR from the *sma-9* versus wild-type RNA-seq. Column labels are in row 1.

Supplementary file 5. Classes of direct targets for SMA-3 and SMA-9. This file lists the direct target genes identified by combined LOA/BETA analysis of RNA-seq and Chromatin immunoprecipitation sequencing (ChIP-seq data), as described in *Figure 3*. Direct targets of SMA-3 alone are in the tab labeled '*Figure 3b*.' Direct targets of SMA-3 and SMA-9 in which both factors promote the target's expression are in the tab labeled '*Figure 3c*.' Direct targets of SMA-3 and SMA-9 in which the two factors have opposite effects on the target's expression are in the tab labeled '*Figure 3d*.' Direct targets of SMA-9 alone in which the factor either promotes or inhibits the target's expression are in the tabs labeled '*Figure 3e*' and '*Figure 3f*,' respectively. For each gene, WormBase GeneID and public gene name are listed, followed by the log fold change and false discovery rate (FDR) from the *sma-3* versus wild-type RNA-seq, followed by the log fold change and FDR from the *sma-9* versus wild-type RNA-seq. Column labels are in row 1.

MDAR checklist

## Data availability

RNAseq data have been deposited in GEO under accession codes GSE266398, GSM8246389, GSM8246390, GSM8246391, GSM8246392, GSM8246393, GSM8246394, GSM8246395, GSM8246396, GSM8246397.

The following datasets were generated:

| Author(s) | Year | Dataset title | Dataset URL | Database and Identifier |
|---|---|---|---|---|
| Vora M, Dietz J, Wing Z, Liu J | 2024 | Genome-wide analysis of Smad and Schnurri transcription factors in *C. elegans* | https://www.ncbi. nlm.nih.gov/geo/ query/acc.cgi?acc= GSE266398 | NCBI Gene Expression Omnibus, GSE266398 |
| Rongo C | 2024 | N2-1, L2 | https://www.ncbi. nlm.nih.gov/geo/ query/acc.cgi?acc= GSM8246389 | NCBI Gene Expression Omnibus, GSM8246389 |

*Continued on next page*

*Continued*

| Author(s) | Year | Dataset title | Dataset URL | Database and Identifier |
|---|---|---|---|---|
| Rongo C | 2024 | N2-2, L2 | https://www.ncbi.nlm.nih.gov/geo/query/acc.cgi?acc=GSM8246390 | NCBI Gene Expression Omnibus, GSM8246390 |
| Rongo C | 2024 | N2-3, L2 | https://www.ncbi.nlm.nih.gov/geo/query/acc.cgi?acc=GSM8246391 | NCBI Gene Expression Omnibus, GSM8246391 |
| Rongo C | 2024 | sma-3(wk30)-1, L2 | https://www.ncbi.nlm.nih.gov/geo/query/acc.cgi?acc=GSM8246392 | NCBI Gene Expression Omnibus, GSM8246392 |
| Rongo C | 2024 | sma-3(wk30)-1, L2 | https://www.ncbi.nlm.nih.gov/geo/query/acc.cgi?acc=GSM8246393 | NCBI Gene Expression Omnibus, GSM8246393 |
| Rongo C | 2024 | sma-3(wk30)-3, L2 | https://www.ncbi.nlm.nih.gov/geo/query/acc.cgi?acc=GSM8246394 | NCBI Gene Expression Omnibus, GSM8246394 |
| Rongo C | 2024 | sma-9(ok1628)-1, L2 | https://www.ncbi.nlm.nih.gov/geo/query/acc.cgi?acc=GSM8246395 | NCBI Gene Expression Omnibus, GSM8246395 |
| Rongo C | 2024 | sma-9(ok1628)-2, L2 | https://www.ncbi.nlm.nih.gov/geo/query/acc.cgi?acc=GSM8246396 | NCBI Gene Expression Omnibus, GSM8246396 |
| Rongo C | 2024 | sma-9(ok1628)-3, L2 | https://www.ncbi.nlm.nih.gov/geo/query/acc.cgi?acc=GSM8246397 | NCBI Gene Expression Omnibus, GSM8246397 |

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
