## [Editor Report · eLife Assessment]

Modulation of BMP signaling affects body size in the nematode *Caenorhabditis elegans*, and this paper examines the effects on *C. elegans* body size brought about by the modulation of BMP signaling. The study provides **valuable** analyses of ChIP-seq and RNA-seq data to understand the function of SMA-3 (Smad) and SMA-9 (Schnurri) in this model. The authors provide **compelling** evidence that the BMP-dependent body size effect could be due to defects in cuticle collagen secretion, a finding of interest to those studying organismal growth and epidermal function.

---

## [Referee Report · Reviewer #1 (Public review)]

Summary:

BMP signaling is, arguably, best known for its role in the dorsoventral patterning, but not in nematodes, where it regulates body size. In their paper, Vora et al. analyze ChIP-Seq and RNA-Seq data to identify direct transcriptional targets of SMA-3 (Smad) and SMA-9 (Schnurri) and understand the respective roles of SMA-3 and SMA-9 in the nematode model *Caenorhabditis elegans*. The Authors use SMA-3 and SMA-9 ChIP-Seq data and RNA-Seq data from SMA-3 and SMA-9 mutants, and bioinformatic analyses to identify the genes directly controlled by these two transcription factors (TFs) and find approximately 350 such targets for each. They show that all SMA-3-controlled targets are positively controlled by SMA-3 binding, while SMA-9-controlled targets can be either up- or downregulated by SMA-9. 129 direct targets were shared by SMA-3 and SMA-9, and, curiously, the expression of 15 of them was activated by SMA-3 but repressed by SMA-9. In case of such opposing effects, the SMA-9 appears to act epistatically to SMA-3. Since genes responsible for cuticle collagen production were eminent among the SMA-3 targets, the Authors focused on trying to understand the body size defect known to be elicited by the modulation of BMP signaling. Vora et al. provide compelling evidence that this defect is likely to be due to problems with the BMP signaling-dependent collagen secretion necessary for cuticle formation.

Strengths:

Vora et al. provide a valuable analysis of ChIP-Seq and RNA-Seq datasets, which will be very useful for the community. They also shed light on the mechanism of the BMP-dependent body size control by identifying SMA-3 target genes regulating cuticle collagen synthesis and by showing that downregulation of these genes affects body size in *C. elegans*.

Weaknesses:

(1) Although the analysis of the SMA-3 and SMA-9 ChIP-Seq and RNA-Seq data is extremely useful, the goal "to untangle the roles of Smad and Schnurri transcription factors in the developing *C. elegans* larva", has not been reached. While the role of SMA-3 as a transcriptional activator appears to be quite straightforward, the function of SMA-9 in the BMP signaling remains obscure.

(2) The Authors clearly show that both TFs can bind independently of each other, however, by using distances between SMA-3 and SMA-9 ChIP peaks, they claim that when the peaks are close these two TFs likely act as complexes. In the absence of proof that SMA-3 and SMA-9 physically interact (e.g. that they co-immunoprecipitate - as they do in *Drosophila*), this is an unfounded claim, which still has to be experimentally substantiated. In the revised version of the manuscript, the authors acknowledge this.

(3) The second part of the results (the collagen story) is loosely connected the first part. dpy-11 encodes an enzyme important for cuticle development, and it is a differentially expressed direct target of SMA-3. dpy-11 can be bound by SMA-9, but it is not affected by this binding according to RNA-Seq. Thus, technically, this part of the paper does not require any information about SMA-9. However, this can likely be improved by addressing the function of the 15 genes, with the opposing mode of regulation by SMA-3 and SMA-9.

Comments on revisions:

In comparison to the first version of the manuscript, the authors have significantly improved the "readability" of the paper, made the Discussion much better, and toned down some of the less supported arguments.

---

## [Author Response]

The following is the authors’ response to the original reviews.

**Reviewer #1 (Public Review):**
Summary:BMP signaling is, arguably, best known for its role in the dorsoventral patterning, but not in nematodes, where it regulates body size. In their paper, Vora et al. analyze ChIP-Seq and RNA-Seq data to identify direct transcriptional targets of SMA-3 (Smad) and SMA-9 (Schnurri) and understand the respective roles of SMA-3 and SMA-9 in the nematode model *Caenorhabditis elegans*. The authors use publicly available SMA-3 and SMA-9 ChIP-Seq data, own RNA-Seq data from SMA-3 and SMA-9 mutants, and bioinformatic analyses to identify the genes directly controlled by these two transcription factors (TFs) and find approximately 350 such targets for each. They show that all SMA-3-controlled targets are positively controlled by SMA-3 binding, while SMA-9-controlled targets can be either up or downregulated by SMA-9. 129 direct targets were shared by SMA-3 and SMA-9, and, curiously, the expression of 15 of them was activated by SMA-3 but repressed by SMA-9. Since genes responsible for cuticle collagen production were eminent among the SMA-3 targets, the authors focused on trying to understand the body size defect known to be elicited by the modulation of BMP signaling. Vora et al. provide compelling evidence that this defect is likely to be due to problems with the BMP signaling-dependent collagen secretion necessary for cuticle formation.

We thank the reviewer for this supportive summary. We would like to clarify the status of the publicly available ChIP-seq data. We generated the GFP tagged SMA-3 and SMA‑9 strains and submitted them to be entered into the queue for ChIP-seq processing by the modENCODE (later modERN) consortium. Thus, the publicly available SMA-3 and SMA-9 ChIP-seq datasets used here were derived from our efforts. Due to the nature of the consortium’s funding, the data were required to be released publicly upon completion. Nevertheless, our current manuscript provides the first comprehensive analysis of these datasets. We have updated the text to clarify this point.

Strengths:Vora et al. provide a valuable analysis of ChIP-Seq and RNA-Seq datasets, which will be very useful for the community. They also shed light on the mechanism of the BMP-dependent body size control by identifying SMA-3 target genes regulating cuticle collagen synthesis and by showing that downregulation of these genes affects body size in *C. elegans*.Weaknesses:(1) Although the analysis of the SMA-3 and SMA-9 ChIP-Seq and RNA-Seq data is extremely useful, the goal "to untangle the roles of Smad and Schnurri transcription factors in the developing *C. elegans* larva", has not been reached. While the role of SMA-3 as a transcriptional activator appears to be quite straightforward, the function of SMA-9 in the BMP signaling remains obscure. The authors write that in SMA-9 mutants, body size is affected, but they do not show any data on the mechanism of this effect.

We thank the reviewer for directing our attention to the lack of clarity about SMA-9’s function. We have revised the text to highlight what this study and others demonstrate about SMA-9’s role in body size. Simply stated, SMA-9 is needed together with SMA-3 to promote the expression of genes involved in one-carbon metabolism, collagens, and chaperones, all of which are required for body size. SMA-3 has additional, SMA-9-independent transcriptional targets, including chaperones and ER secretion factors, that also contribute to body size. Finally, SMA-9 regulates additional targets independent of SMA-3 that likely have a minimal role in body size. We have adjusted Figure 5 with new graphs of the original data to make these points more clear.

(2) The authors clearly show that both TFs can bind independently of each other, however, by using distances between SMA-3 and SMA-9 ChIP peaks, they claim that when the peaks are close these two TFs act as complexes. In the absence of proof that SMA-3 and SMA-9 physically interact (e.g. that they co-immunoprecipitate - as they do in *Drosophila*), this is an unfounded claim, which should either be experimentally substantiated or toned down.

We acknowledge that we have not demonstrated a physical interaction between SMA-3 and SMA-9 through a co-immunoprecipitation, and we have indicated in the text that a formal biochemical demonstration would be required to make this point. Moreover, we toned down the text by stating that our results suggest that either SMA-3 and SMA-9 frequently bind as either subunits in a complex or in close vicinity to each other along the DNA. As the reviewer has indicated, a physical interaction between Smads and Schnurris has been amply demonstrated in other systems. A limitation in these previous studies is that only a small number of target genes were analyzed. Our goal in this study was to determine how widespread this interaction is on a genomic scale. Our analyses demonstrate for the first time that a Schnurri transcription factor has significant numbers of both Smad-dependent and Smad-independent target genes. We have revised the text to clarify this point.

(3) The second part of the paper (the collagen story) is very loosely connected to the first part. dpy-11 encodes an enzyme important for cuticle development, and it is a differentially expressed direct target of SMA-3. dpy-11 can be bound by SMA-9, but it is not affected by this binding according to RNA-Seq. Thus, technically, this part of the paper does not require any information about SMA-9. However, this can likely be improved by addressing the function of the 15 genes, with the opposing mode of regulation by SMA-3 and SMA-9.

We appreciate this suggestion and have clarified in the text how SMA-9 contributes to collagen organization and body size regulation.

(4) The Discussion does not add much to the paper - it simply repeats the results in a more streamlined fashion.

We thank the reviewer for this suggestion. We have added more context to the Discussion.

**Reviewer #2 (Public Review):**
In the present study, Vora et al. elucidated the transcription factors downstream of the BMP pathway components Smad and Schnurri in *C. elegans* and their effects on body size. Using a combination of a broad range of techniques, they compiled a comprehensive list of genome-wide downstream targets of the Smads SMA-3 and SMA-9. They found that both proteins have an overlapping spectrum of transcriptional target sites they control, but also unique ones. Thereby, they also identified genes involved in one-carbon metabolism or the endoplasmic reticulum (ER) secretory pathway. In an elaborate effort, the authors set out to characterize the effects of numerous of these targets on the regulation of body size in vivo as the BMP pathway is involved in this process. Using the reporter ROL-6::wrmScarlet, they further revealed that not only collagen production, as previously shown, but also collagen secretion into the cuticle is controlled by SMA-3 and SMA-9. The data presented by Vora et al. provide in-depth insight into the means by which the BMP pathway regulates body size, thus offering a whole new set of downstream mechanisms that are potentially interesting to a broad field of researchers.The paper is mostly well-researched, and the conclusions are comprehensive and supported by the data presented. However, certain aspects need clarification and potentially extended data.(1) The BMP pathway is active during development and growth. Thus, it is logical that the data shown in the study by Vora et al. is based on L2 worms. However, it raises the question of if and how the pattern of transcriptional targets of SMA-3 and SMA-9 changes with age or in the male tail, where the BMP pathway also has been shown to play a role. Is there any data to shed light on this matter or are there any speculations or hypotheses?

We agree that these are intriguing questions, and we are interested in the roles of transcriptional targets at other developmental stages and in other physiological functions, but these analyses are beyond the scope of the current study.

(2) As it was shown that SMA-3 and SMA-9 potentially act in a complex to regulate the transcription of several genes, it would be interesting to know whether the two interact with each other or if the cooperation is more indirect.

A physical interaction between Smads and Schnurri has been amply demonstrated in other systems. Our goal in this study was not to validate this physical interaction, but to analyze functional interactions on a genome-wide scale.

(3) It would help the understanding of the data even more if the authors could specifically state if there were collagens among the genes regulated by SMA-3 and SMA-9 and which.

We thank the reviewer for this suggestion. col-94 and col-153 were identified as direct targets of both SMA-3 and SMA-9. We noted this in the Discussion.

(4) The data on the role of SMA-3 and SMA-9 in the regulation of the secretion of collagens from the hypodermis is highly intriguing. The authors use ROL-6 as a reporter for the secretion of collagens. Is ROL-6 a target of SMA-9 or SMA-3? Even if this is not the case, the data would gain even more strength if a comparable quantification of the cuticular levels of ROL-6 were shown in Figure 6, and potentially a ratio of cuticular versus hypodermal levels. By that, the levels of secretion versus production can be better appreciated.

We previously showed that rol-6 mRNA levels are reduced in dbl-1 mutants at L2, but RNA-seq analysis did not find enough of a statistically significant change in rol-6 to qualify it as a transcriptional target and total levels of protein are also not significantly reduced in mutants. We added this information in the text.

(5) It is known that the BMP pathway controls several processes besides body size. The discussion would benefit from a broader overview of how the identified genes could contribute to body size. The focus of the study is on collagen production and secretion, but it would be interesting to have some insights into whether and how other identified proteins could play a role or whether they are likely to not be involved here (such as the ones normally associated with lipid metabolism, etc.).

We have added more information to the Discussion.

**Reviewer #1 (Recommendations For The Authors):**
Figure 1 - Figure 3: The authors might want to think about condensing this into two figures.

To avoid confusion with the different workflows, we prefer to keep these as three separate figures.

Figure 1a-b: Measurement unit missing on X.

We added the unit “bps” to these graphs.

Line 244-246: The authors should stress in the Results that they analyzed publicly available ChIP-Seq data, which was not generated by them, - not just by providing a reference to Kudron et al., 2018. As far as I understood, ChIP was performed with an anti-GFP antibody. Please mention this, and specify the information about the vendor and the catalog number in the Methods.

We would like to clarify the status of the publicly available ChIP-seq data. We generated the GFP tagged SMA-3 and SMA‑9 strains and submitted them to be entered into the queue for ChIP-seq processing by the modENCODE (later modERN) consortium. Thus, the publicly available SMA-3 and SMA-9 ChIP-seq datasets used here were derived from our efforts. Due to the nature of the consortium’s funding, the data were required to be released publicly upon completion. Nevertheless, our current manuscript provides the first comprehensive analysis of these datasets. We have clarified these issues in the text. We have also added information regarding the anti-GFP antibody to the Methods.

Line 267-270: The authors should either provide experimental evidence that SMA-3 and SMA-9 form complexes or write something like "significant overlap between SMA-3 and SMA-9 peaks may indicate complex formation between these two transcription factors as shown in *Drosophila*" - but in the absence of proof, this must be a point for the Discussion, not for the Results. Moreover, similar behavior of fat-6 (overlapping ChIP peaks) and nhr-114 (non-overlapping ChIP peaks) in SMA-3 and SMA-9 mutants may be interpreted as a circumstantial argument against SMA-3/SMA-9 complex formation (see Lines 342-348). Importantly, since ChIP-Seq data are available for a wide array of *C. elegans* TFs, it would be very useful to have an estimate of whether SMA-3/SMA-9 peak overlap is significantly higher than the peak overlap between SMA-3 and several other TFs expressed at the same L2 stage.

We have clarified our goals regarding SMA-3 and SMA-9 interactions and softened our conclusions by indicating in the text that a formal biochemical demonstration would be required to demonstrate a physical interaction. Moreover, we toned down the text by stating that our results suggest that either SMA-3 and SMA-9 frequently bind as either subunits in a complex or in close vicinity to each other along the DNA. We have added an analysis of HOT sites to address overlap of binding with other transcription factors. We disagree with the interpretation that transcription factors with non-overlapping sites cannot act together to regulate gene expression; however, nhr-114 also has an overlapping SMA-3 and SMA-9 site, so this point becomes less relevant. We have clarified the categorization of nhr-114 in the text.

Lines 272-292: The authors do not comment on the seemingly quite small overlap between the RNA-Seq and the ChIP-Seq dataset, but I think they should. They have 3205 SMA-3 ChIP peaks and 1867 SMA-3 DEGs, but the amount of directly regulated targets is 367. It is important that the authors provide information on the number of genes to which their peaks have been assigned. Clearly, this will not be one gene per peak, but if it were, this would mean that just 11.5% of bound targets are really affected by the binding. The same number would be 4.7% for the SMA-9 peaks.

We have added a discussion of the discrepancy between binding sites and DEGs. The high number of additional sites classified as non-functional could represent the detection of weak affinity targets that do not have an actual biological purpose. Alternatively, these sites could have an additional role in DBL-1 signaling besides transcriptional regulation of nearby genes, or they could be regulating the expression of target genes at a far enough distance to not be detected by our BETA analysis as per the constraints chosen for the analysis. The difference between total binding sites and those associated with changes in gene expression underscores the importance of combining RNA-seq with ChIP-seq to identify the most biologically relevant targets. And as the reviewer indicated, more than one gene can be assigned to a single neighboring peak.

Lines 294-323: I feel like there is a terminology problem, which makes reading very difficult. The authors use "direct targets" as bound genes with significant expression change, but then run into a problem when the gene is bound by SMA-9 and SMA-3, but significant expression change is only associated with one of the two factors. I am not sure this is consistent with the idea of the SMA3/SMA9 complex. Also, different modalities of the SMA3 and SMA9 effect in 15 cases can be explained by co-factors. Reading would be also simplified if the order of the panels in Figure 3 were different. Currently, the authors start their explanation by referring to the shared SMA-3/SMA-9 targets (Figures 3c-d), and only later come to Figure 3b. In general, the authors should start with a clear explanation of what is on the figure (currently starting on Line 313), otherwise, it is unclear why, if the authors only discuss common targets, it is not just 114+15=129 targets, but more.

We have re-ordered the columns in Figure 3 to match the order discussed in the text. We also incorporated more precise language about regulation by SMA-3 and/or SMA-9 in the text.

Lines 325-355: The chapter has a rather unfortunate name "Mechanisms of integration of SMA-3 and SMA-9 function", although the authors do not provide any mechanism. Using 3 target genes, they show that if the regulatory modality of SMA-3 and SMA-9 is the same (2 examples), there is no difference in the expression of the targets, but if the modalities are opposing (1 example), SMA-9 repressive action is epistatic to the SMA-3 activating action. Can this be generalized? The authors should test all their 15 targets with opposite regulations. Moreover, it seems obvious to ask whether the intermediate phenotype of the double-mutants can be attributed to the action of these 15 genes activated by SMA-3 and repressed by SMA-9. I would suggest testing this by RNAi. I would also suggest renaming the chapter to something better reflecting its content.

We have removed the word “mechanism” from the title of this section. We also performed additional RT-PCR experiments on another 5 targets with opposing directions of regulation. The results from these genes are consistent with the result from C54E4.5, demonstrating that the epistasis of sma-9 is generalizable.

Figure 4b: Why was a two-way ANOVA performed here? With the small number of measurements, I would consider using a non-parametric test.

These data are parametric and the distribution of the data is normal, so we chose to use a parametric test (ANOVA).

Lines 354-355. The authors offer two suggestions for the mechanism of the epistatic action of SMA-9 on SMA-3 in the case of C54E4.5, but this is something for the Discussion. If they want to keep it in the Results they should address this experimentally by performing SMA-3 ChIP-seq in the SMA-9 mutants and SMA-9 ChIP-Seq in the SMA-3 mutants.

We moved these models to the discussion as suggested.

Lines 365-367: "We expect that clusters of genes involved in fatty acid metabolism and innate immunity mediate the physiological functions of BMP signaling in fat storage and pathogen resistance, respectively." - This is pretty confusing since the Authors claim in the previous sentence that regulation of immunity by SMA-9 is TGF-beta independent.

Co-regulation of immunity by BMP signaling and SMA-9 is already known. The novel insight is that SMA-9 may have an additional independent role in immunity. We have clarified the language to address this confusion.

Lines 377, and 380: Please explain in non-*C. elegans*-specific terminology, what rrf-3 and LON-2 are (e.g. write "glypican LON-2" instead of just "LON-2") and add relevant references.

We added information on the proteins encoded by these genes.

Lines 382-384: I am not sure what the Authors mean here by "more limiting".

We substituted the phrase “might have a more prominent requirement in mediating the exaggerated growth defect of a lon-2 mutant”.

Lines 388-392: I found this very confusing. What were these 36 genes? Were these direct targets of SMA-3, SMA-9, or both? Top 36 targets? 36 targets for which mutants are available?

The new Figure 5 clarifies whether target genes are SMA-3-exclusive, SMA-9-exclusive, or co-regulated. The text was also updated for clarity.

Line 397: This is the first time the authors mention dpy-11 but they do not say what it is until later, and they do not say whether it is a target of SMA3/SMA9. Checking Figure 3, I found that it is among the 238 genes bound by both but upregulated only by SMA3. The authors need to explicitly state this - from this point on, they have a section for which SMA-9 appears to be irrelevant.

We added the molecular function of dpy-11 at its first mention. Furthermore, we included the hypothesis that SMA-3 may regulate collagen secretion independently of SMA-9. Our subsequent results with sma-9 mutants disprove this hypothesis.

Line 402: Is ROL-6 a SMA-3/SMA-9 target or just a marker gene?

We previously showed that rol-6 mRNA levels are reduced in dbl-1 mutants at L2, but RNA-seq analysis did not find enough of a statistically significant change in rol-6 to qualify it as a transcriptional target and total levels of protein are also not significantly reduced in mutants. We added this information in the text.

Line 421: I am not sure what "more skeletonized" means.

Replaced with “thinner and skeletonized”

Figure 2b and 2d legends: "Non-target genes nevertheless showing differential expression are indicated with green squares." (l. 581-582 and again l. 588-589) I think should be "Non-direct target genes...".

Changed to “non-direct target genes”

Figure 7 legend: Please indicate the scale bar size in the legend.

Indicated the scale bar size in the legend.

Figure 7: The ER marker is referred to as "ssGFP::KDEL" (in the image and Line 700), however in the text it is called "KDEL::oxGFP" (Line 419). Please use consistent naming.

We fixed the inconsistent naming.

All the experiment suggestions made are optional and can, in principle, be ignored if the authors tone down their claims (for example, the SMA-3/SMA-9 complex formation).
**Reviewer #2 (Recommendations For The Authors):**
(1) As a control: Have the authors found the known regulated genes among the differentially regulated ones?

Previously known target genes such as fat-6 and zip-10 were identified here. We have added this information in the text.

(2) How many repetitions were performed in Figure 4b? I am wondering as the deviation for C54E4.5 is quite large and that makes me worry that the significant differences stated are not robust.

There were two biologically independent collections from which three cDNA syntheses were analyzed using two technical replicates per point.

(3) Lines 333-336: Can you really make this claim that the antagonistic effects seen in the regulation of body size can be correlated with some targets being regulated in the opposite direction? I would assume that the situation is far more complex as SMADs also regulate other processes.

We agree with the reviewer that multiple models could explain this antagonism, and we have added distinct alternatives in the text.

(4) Lines 367-369: Add the respective reference please.

We have added the relevant references.